# Social reputation influences on liking and willingness-to-pay for artworks: A multimethod design investigating choice behavior along with physiological measures and motivational factors

**Blanca T. M. Spee**[1,2]*, **Matthew Pelowski**[1,2], **Jozsef Arato**[2], **Jan Mikuni**[1,3], **Ulrich S. Tran**[1], **Christoph Eisenegger**[1†], **Helmut Leder**[1,2]

1 Faculty of Psychology, Department of Cognition, Emotion, and Methods in Psychology, University of Vienna, Vienna, Austria, 2 Vienna Cognitive Science Hub, University of Vienna, Vienna, Austria, 3 Department of Psychology, Keio University, Keio, Japan

† Deceased.
* blanca.spee@univie.ac.at

**Data Availability Statement:** All data were collected in a manner consistent with ethical

## Abstract

Art, as a prestigious cultural commodity, concerns aesthetic and monetary values, personal tastes, and social reputation in various social contexts—all of which are reflected in choices concerning our liking, or in other contexts, our actual willingness-to-pay for artworks. But, how do these different aspects interact in regard to the concept of social reputation and our private versus social selves, which appear to be essentially intervening, and potentially conflicting, factors driving choice? In our study, we investigated liking and willingness-to-pay choices using—in art research—a novel, forced-choice paradigm. Participants (N = 123) made choices from artwork-triplets presented with opposing artistic quality and monetary value-labeling, thereby creating ambiguous choice situations. Choices were made in either private or in social/public contexts, in which participants were made to believe that either art-pricing or art-making experts were watching their selections. A multi-method design with eye-tracking, neuroendocrinology (testosterone, cortisol), and motivational factors complemented the behavioral choice analysis. Results showed that artworks, of which participants were told were of high artistic value were more often liked and those of high monetary-value received more willingness-to-pay choices. However, while willingness-to-pay was significantly affected by the presumed observation of art-pricing experts, liking selections did not differ between private/public contexts. Liking choices, compared to willingness-to-pay, were also better predicted by eye movement patterns. Whereas, hormone levels had a stronger relation with monetary aspects (willingness-to-pay/ art-pricing expert). This was further confirmed by motivational factors representative for reputation seeking behavior. Our study points to an unexplored terrain highlighting the linkage of social reputation mechanisms and its impact on choice behavior with a ubiquitous commodity, art.

standard for the treatment of human subjects. The data are available in a Figshare repository, accessible via the following DOI: https://doi.org/10.6084/m9.figshare.18865838.v1. The full list of artwork stimuli used is listed in the Supporting Information files.

**Funding:** The author(s) received no specific funding for this work.

**Competing interests:** The authors have declared that no competing interests exist.

## Introduction

If contemporary art were a dialogue, it would be one about values; values that were, values that exist now, and values that—depending on context, personality, and social situation—might dynamically change [1]. Especially with the advent of the scholarly discussions on judgment and taste in the 18th century [2], and increasingly so in the 20th and 21st centuries [3], art has had an interesting place of distinction and social relevance [4]. Today, art has become, more and more, a ubiquitous commodity [5,6] related to a number of values [7], notable among which are art's artistic/aesthetic and economic merits.

These two factors—essentially involving our decisions regarding whether one finds an artwork aesthetically pleasing or preferred and regarding whether one would actually be willing to pay to own a work—are argued to be at the root of many of our responses and uses of art [8–10]. They inform whether we would visit art in a museum, display it on our walls, communicate with our friends about it and how we might cognitively and affectively respond within a given art engagement. At the same time, these factors also raise a number of questions, especially regarding the realization that they do not occur in a vacuum, but rather emerge within a complex context involving backgrounds, motives [11–13], sociocultural habits [14,15], and—in tandem with anecdotal evidence—that they may not always coincide, but may decouple or even contradict [16,17]. Think of the contemporary masterpieces showcased in living rooms of Architectural Digest or the label of wealthy donors affixed in museums, or the works exchanged in the contemporary art auction circuit, signaling one's social and economic prowess. However, to follow a common cynical refrain, one might raise questions of whether their owners actually enjoy them or think of them as good artworks.

Personal preference or liking, is often associated more with formal appearance or the skill of an artist [8,10,18–22]. Setting aside most individuals' lack of purchasing resources to buy a top-line artwork, we may enjoy art while walking through a museum but do not want to own it [23,24]. Alternatively, we may love pieces purchased at a flea market or poster copies of famous artworks, to simply have and see them, daily, in our own spaces. We may of course also both like and be willing to buy the same art. This suggests a complex interplay of value factors and their effect on the choices types—how do we make these choices; how do they interact?—which however is still rather undefined in empirical and theoretical research [9].

One explanation for our liking and willingness-to-pay choices—and especially their combination—is the differential relation to social reputation [25,26], which may directly intervene at the intersection of artistic and monetary values and personal choice. Social reputation can be defined, in general, as the respect, esteem, or prestige that a person (or object) has in the eyes of others (i.e., expression of social status, [27,28]). As social beings, it is commonly acknowledged that humans routinely act with social reputation in mind as a key factor in driving their behaviors [3,12,20,26]. We want to have esteem among others, be seen as fitting in, or separate us from certain peer groups [29]. A major way of doing this is to broadcast who we socially are, or wish to be, via our actions and choices—a manifestation of our identities often attributed to a 'social-self'—and which may or may not overlap with our more personal choices and values, if made in private.

In the domain of art, social reputation has mainly been discussed in the fields of contemporary art education, social economics [30,31], as well as in sociological and social psychological research (e.g., [32–35]), where it is commonly acknowledged that art as object or activity (viewing, owning, and also making art) does routinely carry high social reputation or act as vehicle for communicating certain aspects [16,36]. Art purchasing, as in the above examples, considering art as investment and ownership, has a long history of broadcasting reputational

potential through socioeconomic status, [16,37, see also 30,31,34]. A purchase is, in most cases, a public act, as is communication to others about what something costs.

Liking art, although often perceived as more personal or idiosyncratic, is also associated with social reputation [9,20,21]. In sociological surveys, spending time viewing art or choosing to visit a museum is often argued to be driven by desires to join or distinguish oneself from certain social groups (e.g., [9,15]). Affiliation with the arts has been connected to higher education, status, and personal wealth [9,30,31]. Personal choices displaying 'good' art may also act as a means of broadcasting abilities. Discussions of aesthetic sensitivity or taste [9,38–40]—not to mention more contemporary acts such as giving 'likes' to images or posting attractive selfies on social media [41,42]—have long held an unavoidable overlap with more general discussions of skill, merit, and socioeconomic status [43,44].

In turn, art appraisals have been shown to be changeable by manipulating social context. Informing individuals that paintings are from a prestigious museum (compared to computer-generated images) increased relative liking ratings [45]. Similar impacts, on liking and valuation, has been found by informing individuals that art is by a prestigious artist versus one of their students, is an original versus a copy [46], is eventually a fake [47], or even that a prestigious company sponsored the study [48]. Telling individuals that artworks were liked by one's peers or by art experts, or disliked by socially undesirable others, also modulate liking versus ratings made without social context [9] (see [49] for similar study with popular music; see also for further reading [50–52]).

These arguments, on the surface, would seem to be rather intuitive—our selections and purchases are also modulated by our social environment. However, an interesting implication emerging from these studies is that such social impacts may be rather 'domain specific'—relating especially to what sort of social context and communication one thinks is relevant—which present a compelling suggestion for studying the interaction of these choices.

In a recent paper relating appraisal changes to in- and out-groups' rating information [9], for example, it was found that using a monetary prime (i.e., telling individuals that an artwork was particularly cheap or expensive) had much smaller, and nearly negligible, impacts on liking. Similarly, Newman, Diesendruck and Bloom [46] found that, when asking participants to evaluate the market price of a painting, ratings could be changed by introducing aspects used in such valuations—provenance, scarcity, resources, or effort.

However, value was not impacted by raising more personal preference or esteem information when owners had themselves elevated or downgraded an item from 'art' to 'not-art' status. Detotto and colleagues [37] asked visitors in an art exhibition about their interest in paying to preserve and publicly display art (in this case via the municipality's taxes) and found that willingness-to-pay could be modulated by raising a related factor: whether art reflected the cultural heritage/identity of the location or art made in another country by the same artist.

In one of the only studies to actually combine both aesthetic/artistic and monetary values, Kruger and colleagues [17] found that when participants were informed about a factor that could potentially touch both—the amount of time required to make a piece and "all else being equal," (p. 92)—could modulate both liking and price valuations. These modulations occurred presumably due to the factor's potential as a proxy for quality for both values.

The above relationships have also been shown to be modulated by individual differences regarding motivations or relative importance (to their social or personal self) of making one or the other choice. For example, Kirk and colleagues [48] showed that while studying sponsorship from companies, liking ratings were less modulated in art experts, for whom artistic aspects were probably of higher importance and who did not default to provided value proxies. On the contrary, individuals, who are art laymen and along other personality traits [32,33],

have shown to be more vulnerable to reputation influences, and align with believed experts' opinions, value systems, and preferences (see [9,46–48]).

These findings, especially when considering the interaction of liking and willingness-to-pay assessments might then suggest that it is the specific combination of social context—whether choices of one or the other are made in public or private—and also the relative importance of the social self in these domains, that may drive one or both determinations. On the one hand, an individual believes they are operating in a social domain (either intuitively or due to a study design) that puts a premium on price, then they give this aspect preference in their decisions. This may occur in tandem with more awareness of and susceptibility to the latent related social information, and perhaps with their choices coming at the expense of liking or aesthetic factors. On the other hand, if an individual believes they are operating in a domain putting a premium on taste or artistic appearance, they may focus on and be impacted by these aesthetic features and especially related social information. This could explain purchasing art that one does not like, liking art regardless of the cost, or the impact on choices along the relative importance in certain social domains. Depending on the person or domain, the relative impact of the social context can lead to large differences in the extent to which certain decisions are relevant or default to the information provided. It may well be that liking ratings are for many, more stable and less susceptible to social setting, whereas price valuations are more at risk. Beckert and Rössel [16], looking at economic aspects of the contemporary art market make this claim, that even most individuals who do desire to buy art face a problem of fundamental uncertainty, because of the difficulty in determining value; this leads them often to default to experts in the art field [44]. Similarly, sociological studies of art interest also suggest broad combinations of relative interest in art's economic and/or aesthetic importance, with interactions and tastes largely driven by one or both factors [25].

In sum, this perspective of social reputation suggests that researchers might consider the combination of these factors, assessing both liking and willingness-to-pay, as well as the specific social context, to better tease out their relationships. However, to date, this has not been done. Although again theoretical and empirical investigations have linked art values to socio-cultural training [25,30,31,34], habits [15], and corresponding choice behavior and appreciation [21], and have discussed them as indicative for social reputation effects [e.g., 16,36], there is no empirical research yet directly investigating the relationship between these factors. There are also very few studies actually empirically exploring willingness-to-pay or relationships with pricing information (see for further reading [9] and for non/art domains, e.g., [53–55]), and no study has interlinked, empirically, both liking and willingness-to-pay choices with the manipulation of a social reputation context itself.

## The present study

In this paper, we studied individuals' selections of artworks, regarding artistic (associated with liking) and monetary (associated with willingness-to-pay) values, and how the two choice types (liking, willingness-to-pay) differed due to manipulation of social reputation. We tested our main hypothesis, that social reputation influences art evaluation in terms of choice behavior, by employing a novel forced choice design. We operationalized our within- and between-subject variables along the different aspects of social reputation discussed above.

We used images of real artworks of human artists, which was also communicated to the participants. We labeled the artworks along artistic and monetary values. We operationalized this by presenting in each trial triplets of three similar artworks (see Fig 1 for stimulus example and study design). These were presented together with labels explained as indicating ascending order of levels of artistry (i.e., artworks rated as artistically superior by art experts) with

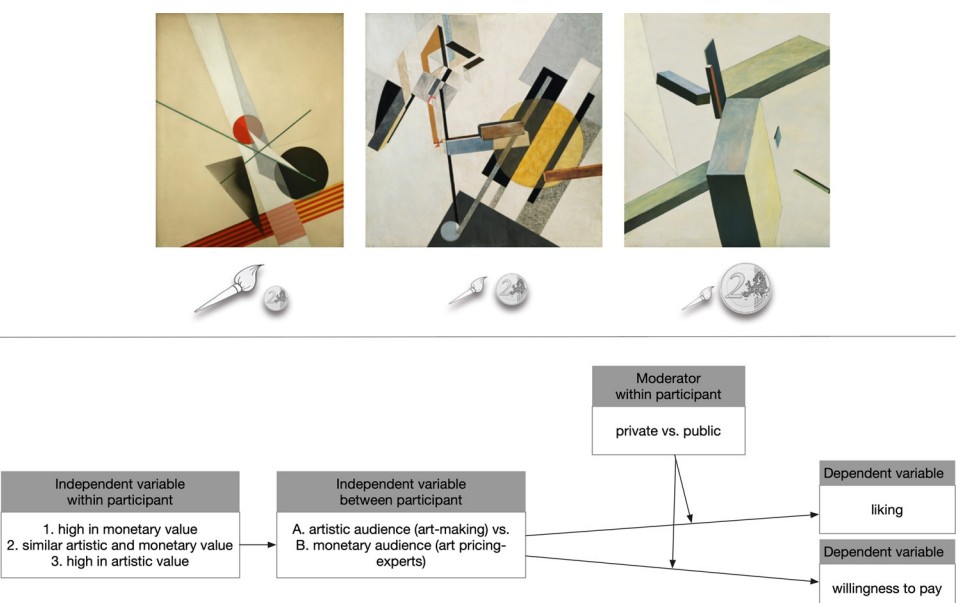

**Fig 1. Example of a stimulus-set including labelling and study design.** Top: From left to right, the artwork's artistic value rose, and from right to left, the monetary value increased (e.g., in the between-group art-making experts, the left artwork is considered a pro-artistic choice). Pictograms represented artistic and the monetary values (by the size of the brush/coin) and were counterbalanced. In addition, the pictograms under each image were also counterbalanced left/ right between participants. The artist of the middle and right artworks is by El Lazar Markovich Lissitzky (known as El Lissitzky) and the left one by László Moholy-Nagy. Copyright information: Shown artworks are in the public domain in its country of origin and other countries and areas where the copyright term is the author's life plus 70 years. Bottom: Modelled study design, see Methods for full description.

likewise descending (opposing) levels of monetary value (according to auction house experts). Through these opposing value labels, we tested whether choice behavior represents the associations discussed above: that is, art of high artistic value is more often chosen to be liked, and participants are more often willing to pay for art of high monetary value (see Stage I-II in Fig 2). In addition, we provided a middle (neutral) position as a choice option in our study design (see Methods for study design). This served to give participants the opportunity to also choose a fallback option in which monetary and artistic value are equal according to the experts.

These actions were also conducted in differing social reputation contexts (see Stage III-V in Fig 2). To investigate this influence, we compared between-group two condition in which we asked participants to choose (forced choice) one artwork they liked most and one they were most willing to pay for in an unobserved block (private) as well as in a public block. In latter, participants were made to believe to be observed by either art-pricing or art-making experts (see Methods for full description).

Our sub-hypotheses regarding behavioral choice behavior were: *1.a If the audience in the public setting are art-making experts, we would expect that participants will like high artistic value artworks more often compared to the private setting.* Likewise, we expected *1.b if the audience in the public setting are art-pricing experts, then participants will be willing to pay more often for artworks assigned to be high in monetary value compared to their choices in the private context.* We further hypothesized, *1.c, that choices remain unaffected if the audience was not relevant to the respective value type.* Specifically, this means, that if the art-pricing experts are watching, liking choices will not change from private to public; and, willingness-to-pay choices will remain the same when art-making experts are observing. A visual summary of all hypotheses and measurements is given in Fig 2.

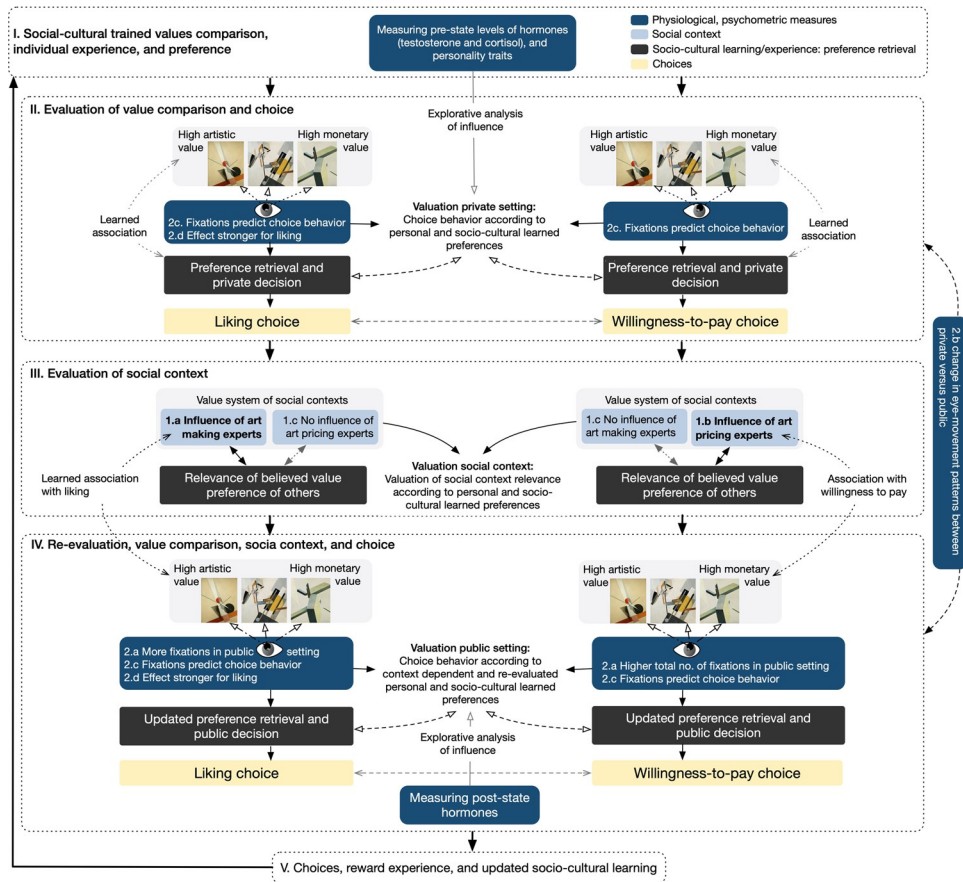

**Fig 2. Visualization of the hypotheses and assumptions for behavioral and implicit measures.** Development of choice behavior described along five evaluation processing stages. Stages I-II: Learned socio-cultural choice behavior and associations; private context choice behavior in stage II (hypotheses 2.c-2-d). Stages III-IV include social context and influence on choice types (hypotheses 1.a-1.c; 2.a-2.d); stage V feed back into the evaluation process and updates personal experience, socio-cultural values, individual experience, and preferences updating stage I. Stages include stimuli-sets, physiological measures (in dark blue), influence of socio-cultural learned behavior (in dark gray), choice types (in yellow), and social context is presented in stage III (light blue).

## Physiological and other measurements

In addition, and as we are studying social influences within a contextual framing situation, this situation itself could influence choices and behavioral measurement and may not reveal potential hidden processes (e.g., choosing a different artwork due to social influence but not because the person likes it). Hence, behavioral choices alone may not suffice to investigate this topic. Additional implicit physiological and scales of motivational factors used in studies investigation hormonal analysis [56–61] could provide crucial information regarding this complex factor interlinkage and choice behavior. We therefore further employed two physiological measures: first, eye-tracking to analyze eye-movement patterns with regression modelling along choice behavior, a method which has been widely used in art research investigating personal appreciation and liking [62–67]. Second, exploratorily, neuroendocrinological hormonal measures, which we report and discuss along descriptive statistics, and which have been discussed within social economic studies along with social reputation effects [68–78]. Additionally, to support our hormonal analysis, we also added scales measuring motivational factors

associated with hormonal associated reputation seeking behavior (see for latest applied study, [60,61,79]) using regressions analysis to predict choices.

**Eye-tracking.**   The assessment of fixations and eye-movement patterns represents an established research method in art studies. Theoretical models and empirical studies in visual art (e.g., [62–65], see for review [10]) indicate that perceptual processing begins from the first moment of visual input and is constantly updated through active eye-movements and fixations on areas of the artwork viewed. Studies have resoundingly shown that individuals spend more time looking at artworks which they find aesthetically appealing and which they liked most (in terms of total fixation, see, e.g., [62,64–67]; or along longer fixation, see [63,80]). Moreover, participants appear to fixate the preferred image when reaching the decision-moment [66–67].

We tested whether participants also fixated the artwork image they choose as liked or that they are willing to pay for. One general assumption was that participants would look around less in a private context, whereas in a public context social influence would distract participants. The first two eye-tracking hypotheses are therefore *2.a a higher total number of fixations in the public condition* (more comprehensive exploration of all artworks and value cues) and *2. b. a change in eye-movement patterns between the private versus public condition* (however, we keep the directionality, how often, and at which image they fixate open).

In accordance with previous findings [60,66–67,80], we expected, *2.c that both*, *the total number of fixations* (artworks where participants look the most) *and the last fixation predict choice behavior overall conditions.* Moreover, relating to the personal connotation of liking discussed in the Introduction, we further hypothesized *2.d that the number of fixations would predict liking choices* (a more personal value and decision) *stronger than willingness-to-pay choices* (monetary values are stronger influenced by other culture and economy matters, potentially irritating eye-movements—e.g., more looking around).

**Neuroendocrinology.**   At a neuroendocrinological level, hormones (e.g., testosterone or cortisol) and neurotransmitters (e.g., dopamine), are necessary to translate external contextual and internal bodily information into brain activation patterns [68]. Regarding social reputation and social status aspects, Eisenegger and colleagues [58,59,69] accentuated the role of the steroid hormone testosterone as an important contributor in achieving and maintaining social status and reputation. Testosterone seems to be a rudimentary influencing component behind the motives of status gathering and keeping, leading to cooperative, fair, or reputation-seeking behavior [58,59,70–72]. However, this output of testosterone functioning is deeply connected to cortisol (stress) levels, a steroid hormone of the glucocorticoid family [73–76]. Testosterone is only strongly associated with status seeking when cortisol levels are low. However, testosterone in interaction with high cortisol levels has been shown to block status-seeking behavior. This interconnection of both hormones is summarized in the dual-hormone hypothesis (see for further reading [73,76]).

Although art research has undergone several epistemological advances in the last years considering aspects of neuroscience [81], the study of neuromodulations is very sparse [82]. To date, there are only very few neuroendocrinological studies in the art field, mainly focusing on stress reduction through art measured along with cortisol levels [77,78]. Our study is probably the first to look at this specific kind of hormonal interaction (testosterone, cortisol). We measured testosterone and cortisol levels (especially cortisol change, pre-/post levels, i.e., hormonal change after psychological stress, see [68,75]) and analyzed it in respect to choice behavior. Based on prior studies in social economics and neuroendocrinology [58,69–76], we expected that participants high in testosterone and low in cortisol would try to keep their general (socio-culturally acquired) reputation, and show high liking for high artistry and are more often willing to pay for high monetary valuable art (along socio-cultural trained habits, see Stage I-II in Fig 2). In the public condition, however, we anticipated two potential strategies

that might be revealed by this line of evidence: in case participants agree with the audience value system, we would expect a strengthened effect of the private behavior, meaning following socio-cultural learned behavior (see Fig 2). Alternatively, if participant disagree with the audience, they might be inclined to go against the believed value systems of the audience and show different choices between private and public in controversial ways. As this is the first study combining these measures and design, we discuss our explorative results in the Discussion.

**Scales of motivational factors.** Last, we also included measures focusing on motivational factors: the behavioral activation (BAS) and inhibition (BIS) questionnaire suggested to measure motivational systems or traits [60,61,79, hereafter BIS/BAS scale] and the Liebowitz Social Anxiety Scale (LSAS-Anxiety Scale, [83]). We included both measures because both represent motivational factors which have been associated with hormonal measures in different fields of research before [6–61,84,85]. The BIS/BAS scale has been significantly correlated with high testosterone levels and status-seeking behavior respecting stress situations and is commonly used in neuropsychopharmacological experiments using testosterone application and used as control measure [60,61,79]. We further included the BIS/BAS scale because it is in accordance with the dual-hormone hypothesis discussed above (see [73,74,76]). To find potential effects of personality and experience of being more prone to get irritated by the public condition, we also included the LSAS-Anxiety Scale [83]. The LSAS-scale has been applied in conjunction mainly with cortisol measures as well as representing social anxiety and social approach avoidance [84,85]. To date, there is no study in art research that has applied the BIS/BAS scale or any anxiety scale within such kind of paradigm. Thus, based on other studies in social economy [e.g., 58–61], we assume, that there may be a positive association with behavioral activation scale, reputation seeking behavior (more prone socio-cultural learned behavior and agree with audience's value system stronger in the public condition, see Stage III-IV in Fig 2), and choice behavior. Generally, employing personality measures might provide additional measures that support the explorative neuroendocrinological data and for future research.

## Materials and method

### Participants

The study involved a final sample of 123 participants ($M_{age}$ = 21.7, SD = 2.9, age-range = 18 to 30; 52.85% female; between-group $n$ = 61 art-making experts and $n$ = 62 art pricing experts). All were students at the University of Vienna. The final sample was derived from an initial collection of 149 participants with 26 participants excluded due to issues with the eye-tracking application. Note that this sample was largely a convenience sample that represented the maximum number of participants that could be assessed within our given budget. Due to the novelty of the design and the exploratory nature of the study, no a priori power analysis was conducted [86]. However, we did conduct a sensitivity power analyses on main results (see Results below).

Further, for the analysis of the saliva samples, only male participants were tested and analyzed ($n$ = 58). Seven male participants from the final total sample had to be excluded from the saliva analysis because the amount of saliva given was insufficient. One person was excluded because of excessively high testosterone levels (~10 times higher than average, despite double analysis of the saliva sample; the cause could not be determined), leaving a final sample of $n$ = 50 for the hormone analysis. The decision to include only male participants for the saliva analysis was due to our desire to omit conflating issues regarding potential variations of hormone levels due to the use of different contraceptives and monthly cycles of female participants. This decision to split the sample further admittedly led to a rather underpowered (in post-hoc sensitivity power analyses) result (i.e., where 50+ participants per between group and

in total over 100 participants would be ideal [69]). Nevertheless, due to the time and costs involved for conducting the study, it was not feasible to test even more participants at that time, which could also not be covered by the budget.

All participants had normal or corrected-to-normal vision (below 1.2 diopters). All participants were art-novices (mainly psychology students) and had no education in fine arts, art history, or other related disciplines dealing with art. Participants were informed about the basic procedure and provided informed consent. Course credits were given after finishing the experiment. The study was conducted in accordance with the standards of ethical principles regarding human experimentation and was approved by the ethical commission of the University of Vienna (reference number ethics committee 00256).

## Stimuli

The study used 48 images of abstract paintings made from about 1960 onward (see Fig 1 for an example; S1 Table in Supplementary Information includes a full list of all artworks). Abstract art was selected to minimize potential confounding issues (e.g., personal memories, associations) from mimetic content and, due to the general, well-documented lack of familiarity, interest, and even appreciation for abstract art from our participant base [19,40,73–75], we expected that participants would find it plausible that abstract contemporary art might well vary in both artistic and monetary axes. Whereas representational paintings might be expected to have a higher, positive correlation between price and artistic quality, or, as historical objects, might not be expected to have especially low prices. We grouped the artworks into triplets. Artworks within a set were chosen to be highly similar in style, color, composition, and often painted by the same artists (Fig 1, see full list of artworts and triplets in S1 Table in Supplementary Information). This matching was important to reduce potential low-level visual feature influence on judgement behavior (like complexity, color usage, etc., [87,88]). We further randomized the location of the artworks within one set (i.e., appearing on the left, right, or in the middle position) between-participants. Furthermore, all artworks had the same height (500 px), but different width so as not to distort the picture content. The regions of interest for the eye-tracking analysis were controlled for each triplet individually to cover the entire area to the outer edge of each of the three artworks. We used pictograms representing artistic and the monetary values (by the size of the brush/coin), which was also explained in the test trials before the experiment started. The pictograms were counterbalanced between-participants resulting that the artistic/monetary high levels exchanged sides and artworks. Also, each pictogram set underneath each image was counterbalanced left/right.

## Procedure

Participants were first welcomed to the lab and asked to sign the informed consent. We then determined the dominant eye by asking participants to focus on an object while alternately covering the other eye. Afterwards, the movement of the dominant eye was calibrated with EyeLink 1000 (SR Research Ltd., Mississauga, Ontario, Canada). Calibration was repeated for each of the two blocks. Male participants were then asked to wait 20 minutes before taking the first saliva samples. Female participants started directly with the experiment. All further instructions were given on the computer screen. Participants read a cover story in which the composition of the triple sets was explained as follows. They were told that these triplets were composed in cooperation with art-making experts (Vienna Academy of Art), art-pricing experts (from a fictious 'Auction house Wittelsburg'), and the research team at the University of Vienna to study the difference of art-values (artistic/monetary). The art-making experts highly valued one of the artworks in the set for its superior formal or artistic and aesthetic

quality (hereafter artistic value). However, this artwork was also less expensive based on a recent purchase price estimated by the art-pricing experts. Another artwork, the opposite artwork in the set, had recently sold at the auction for a high monetary amount, but, at the same time, was argued by art-making experts to be artistically not valuable. The middle paintings in each set had a moderate and similar value in artistic and monetary value and quality (see Supplementary Information S1 File German version and S2 File English version of full cover story).

Part of the cover story was also devoted to explaining the private versus public paradigm. The participants had to perform the procedure two times: once in a private setting and once in a public setting. In the latter block a recording was taken for the respective expert group. Here, the participants were divided into two groups (between-group variable): one group was observed in the public condition by art-making experts, the other by art-pricing experts. To create this public session, the experimenter, without otherwise talking to the participant, installed and switched on (i.e., with visible red light) a camera behind the participants. The order of the sessions was counterbalanced between participants. In case the public condition was the first, the camera was de-installed for the private session. Camera installation/de-installation took about 1 minute, and we confirmed via eye contact, that participants were aware of the public and private conditions.

Before beginning the main procedure, participants also completed three practice trials with paintings from the 48-item set (trial stimuli were excluded from analysis and not used in the experimental trials), to ensure that participants were fully comfortable with the procedure. First, participants were presented with a fixation cross in the middle of the screen (to ensure proper eye-tracking recordings). When fixation succeeded the cross was replaced by one of the three-artwork sets, with accompanying artistic/monetary value information. Participants had a free-viewing time for 20 seconds. Afterwards they had to make two choices. They had to choose one artwork out the triple-set they *liked* most and one for which they were *willing to pay (hereafter* 'wtp') (in case personal monetary wealth would not be an issue) via a keypress. The order of the two choices was randomized per trial. No time limit was set for making the choices.

Upon answering both questions, the procedure was repeated for a total of thirteen trials. The thirteen trials were again repeated in both within-subject blocks (private and public), where participants could choose the same or a different artwork within each of the thirteen triplets. Despite the background story and camera setup, no actual recording was taken. Participants were informed after the experiment that the recording as well as the name of the auction house were made-up.

## Physiological measures: Eye-tracking

The eye-tracking data was recorded with EyeLink 1000 desktop mounted eye tracker (SR Research Ltd., Mississauga, Ontario, Canada), sampling at 1000 Hz. EyeLink 1000 provides two pupil tracking algorithms: centroid and ellipse fitting. The centroid mode was used, tracking the center of the threshold pupil using a center of mass algorithm. We calibrated the apparatus via 9-point calibration procedure with the dominant eye. A chin and forehead rest stabilized participants' head positions and minimized movements during the eye-movement recording. All participants sat at a desk in the, dimly lit, laboratory (background luminance about 500 lux), 60 cm away from the monitor, and viewed the artwork sets on an LCD monitor Samsung SyncMaster 2443BW, with a resolution of 2,400; 1,920; 1,200 pixels and a screen refresh rate of 60 Hz. The experiment was controlled by Experiment Builder Software Version 1.10.1630 (SR Research Ltd., Mississauga, Ontario, Canada) on a Windows PC. Before the

start of each trial, participants were required to look at a fixation cross presented in the middle of the screen to trigger the trial start. If fixation failed within 10 seconds, a 9-point re-calibration and validation was performed.

### Physiological measures: Neuroendocrinology

Saliva samples were taken from the male participants after a waiting period of 20 minutes and before the experiment started. The second samples were taken 20 minutes after the experiment. During the waiting phases (for pre- and post-sample), participants were instructed to avoid any arousing activity—neither physically nor mentally—but to sit still. Use of electronic devices were forbidden during waiting phases, but participants could read some unexciting newspapers. In total, four saliva samples were taken: two pre-samples and two post experiment for analysis of testosterone and cortisol.

### Demographics and scales of motivational factors

About one week before conducting the actual study, participants completed an online survey answering: (a) behavioral inhibition system/behavioral activation system (BIS/BAS) questionnaires [79], (b) a social anxiety scale (LSAS; [83]), (c) general questions about art expertise [89], and (d) standard demographic data was queried, such as gender, age, etcetera. The participants had to provide a self-chosen code (random code with 4 letter and 2 digits), to link the data of the survey to the data gathered during the experimental sessions. The codes were only known by one of the experimenters.

## Results

### Measures and statistical analysis

Regarding the analysis of art preference choices, we focused on two main aspects. First, we examined which painting participants chose in the different conditions. As the value-order was randomized between-groups, we set fixed values for the different values of choices for all participants: 1 = high artistic value, 2 = neutral, and 3 = high monetary value. Despite the discrete (1-2-3) choice options, once averaged across trials, the responses were normally distributed. This was confirmed by a Shapiro-Wilk test, which found no deviation from normality (liking: $W = .985$, $p = .197$, wtp: $W = .991$, $p = .591$).

Second, we assessed the change in choices between the within-group condition private versus public. We therefore subtracted the value in the private condition from the public condition. If participants did not change their opinion, the value would be zero (see Fig 4 as an example). If one participant chose, in the public condition, a high artistic value artwork (value = 1) and, in the private condition, a high monetary value artwork (value = 3), then the change value would be -2. Consequently, a negative value indicated that the participant chose higher artistic value artworks in the public condition. However, if the participant chose a high monetary value artwork in the public condition (value = 3), and high artistic in private (value = 1), a positive value—plus 2—resulted. In such a case, the interpretation of the value is that participants chose higher monetary value artworks in the public condition. In summary, if the value deviated from zero, there was a change in choice behavior: a negative value means more artistic art, and a positive value means more monetary valuable art, was chosen in the public compared to the private condition. Again, the responses were averaged across trials and the variable was interval scaled. Besides 95% confidence intervals (CI), we additionally report statistical effect sizes for $t$ tests with Cohen's $d$ [90], for ANOVAs with eta squared ($\eta^2$); for multiple regressions analyses $r^2$.

We performed sensitivity power analysis for our main comparisons with $\alpha = .05$ and 80% power in G*Power [91]. For the paired $t$-test comparing liking and wtp choices and fixations of all participants, we found a critical minimum Cohen's $d$ of 0.22 ($N = 123$). For the paired comparison of public vs private condition, the minimum value was $d = .361$ for art-pricing experts ($n = 62$) and $d = .365$ for art-making experts ($n = 61$). The mixed within-between ANOVA revealed a critical effect size of $\eta^2 = .061$ for the within group factor and $\eta^2 = .054$ for the between group factor (using the empirical correlation of $r \approx .75$ between the repeated measurements in public vs. private). Finally, for the within-between interaction a critical value of $\eta^2 = .008$ was found. The Cohen's f values returned by G*Power were transformed to eta-squared according to $\eta^2 = f^2 / (1 + f^2)$ ([76], see also the note on ideal sample size for the hormonal analyses above).

Furthermore, for the hormonal analysis participants were clustered in specific hormone-level constellation clusters. The rationale for adding the hormonal measures not as continuous variable but for clustering is based, on the one hand, on the extensive theory of dual-hormone analysis [74–76], which states that testosterone is mediated by cortisol along specific patterns. If the patterns of hormone level constellations can be confirmed by the theory and the present data, clustering is performed using GaussianMixture function of the scikit-learn library [92]. On the other hand, the theory [74–76] predicts that an interaction between testosterone and cortisol influences behavior. Our behavioral results are also based on an interaction, leading to analyzing the interaction of two interactions for which our sample size is too small (since already a threefold interaction can require four times as much data as a twofold interaction; see for further reading, [93]).

## Behavioral results—Artwork choices in liking and willingness-to-pay

The descriptive analysis of liking and willingness to pay (wtp) choices for both the between-group factor audience type (art-making vs. art-pricing experts) and the within-group factor (private vs. public) is shown in Table 1. Per group and round, the average value was determined from 13 choices (13 stimulus sets). Even though many of the values appear to tend towards the middle, artistically valuable artworks received more liking choices, which is visible in that both *means* were below the value 2 (see Table 1, Fig 3).

To exclude a potential effect of position and also to study if participants actually chose mostly the middle position, we also calculated the average choice percentage for the three locations (see S1A Fig in Supplementary Information) and the average choice percentage for the three stimulus types (see S1B Fig in Supplementary Information). Both analyses revealed that participants choose all three positions/stimulus types nearly similarly (around 33.3%) on average with a slight avoidance of the middle/neutral position.

**Table 1. Descriptive analysis liking and willingness-to-pay (wtp) choices for all conditions.**

| Choices | within-group factor | Private | | Public | |
|---------|---------------------|---------|----|--------|----|
| | between-group factor | M | SD | M | SD |
| liking | Art-pricing experts | 1.94 | 0.24 | 1.93 | 0.25 |
| | Art-making experts | 1.96 | 0.20 | 1.94 | 0.21 |
| wtp | Art-pricing experts | 1.99 | 0.30 | 2.05 | 0.32 |
| | Art-making experts | 2.03 | 0.27 | 2.00 | 0.26 |

A value around 1 represents a choice for high artistic valuable artworks, 2 a neutral choice, and 3 a high monetary choice.

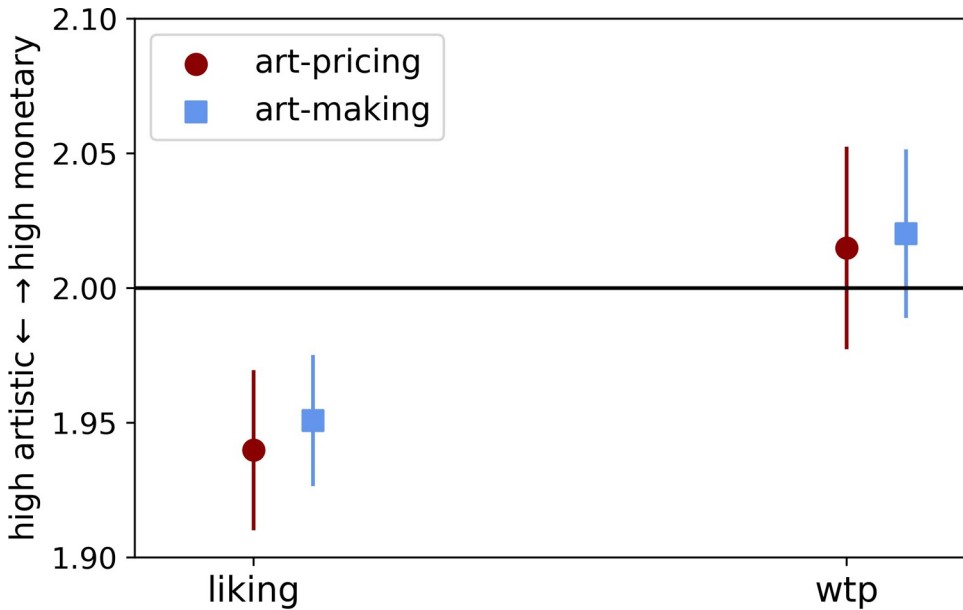

**Fig 3. Descriptive results of liking and willingness-to-pay (wtp) choices between-group conditions.** Error bars represent standard error of the mean (see S2 Fig in Supplementary Information for separation in all condition).

To test hypotheses 1.a and 1.b, we analyzed if there was a general effect of audience type, we applied a two-way mixed ANOVA, with audience type as a between-group factor, choice type (liking, wtp) as a within-group factor, and the actual image choice (i.e., which artwork of the

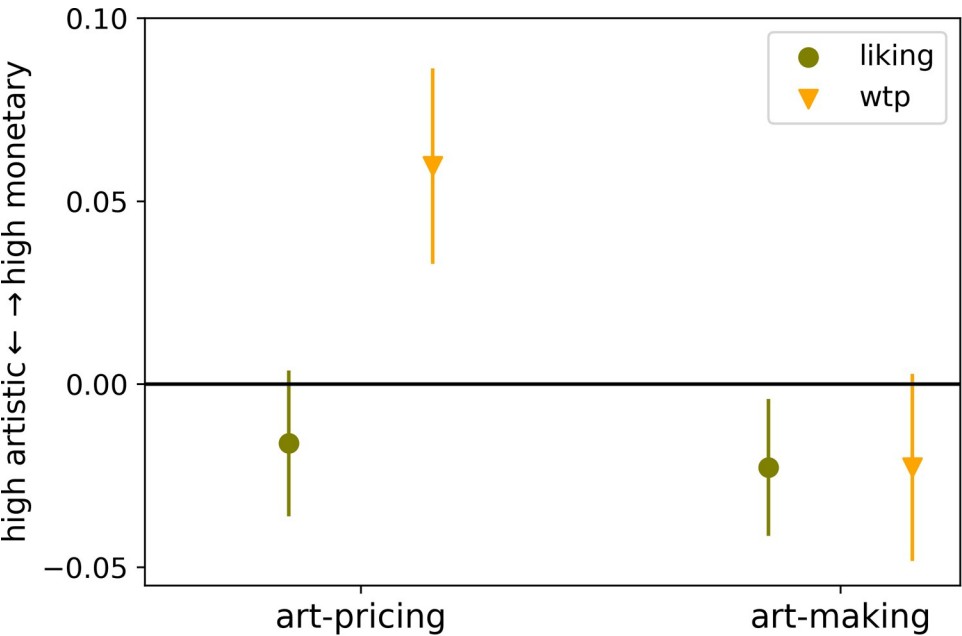

**Fig 4. Change in choice behavior from between the two audience conditions and for each choice type liking and willingness-to-pay (wtp).** The y-axis describes the direction of value from high artistic to high monetary value. Values at the zero line mean no change in decision behavior; negative values mean participants chose higher artistically valuable artworks in the public condition; positive values mean participants choose more high monetary valuable artworks in the public condition. Error-bars represent standard error of the mean.

three in the set the participant picked) as the dependent variable. The analysis revealed no main effect between the two audience groups ($F(1, 121) = .044$, $p = .838$, $\eta^2 = 0.00$) and no interaction ($F(1, 121) = 0.021$, $p = .774$, $\eta^2 = 0.00$). There was, however, a main effect in image choice between liking and wtp ($F(1, 121) = 13.442$, $p < .001$, $\eta^2 = 0.10$), in that participants in both audience groups liked the high artistic value artworks more, and were more wtp for high monetary paintings.

Testing hypothesis 1.c, our second analysis concerned the change in choice behavior between public and private (see Fig 4). A *t* test revealed no significant change in public vs. private in liking in neither audience group (art-pricing experts: $t(61) = -0.81$, $p = 0.422$, 95% CI [-0.06, 0.02], $d = 0.04$; art-making experts: $t(60) = -1.21$, $p = 0.230$, 95% CI [-0.06, 0.01], $d = 0.155$). There was a significant change towards high monetary value from private to public in wtp choices and with audience type art-pricing experts ($t(61) = 2.23$, $p = 0.029$, 95% CI [0.01, 0.11], $d = 0.28$). However, we found no significant change from private to public in wtp, when art-making experts were watching ($t(61) = -0.89$, $p = 0.378$, 95% CI [-0.07, 0.03], $d = 0.113$).

Further, a paired *t* test revealed a main effect between liking and wtp choices with the audience art-pricing experts ($t(61) = -2.49$, $p = .015$, 95% CI [-0.14, -0.01], $d = 0.408$) showing that participants were more wtp for high monetary value artworks and middle to high artistic value artworks were more often chosen to be liked (see Fig 4). On the other hand, as expected (hypotheses 1.c), a paired *t* test revealed no significant change in choice behavior for this group between liking vs. wtp for art ($t(60) = 0.00$, $p = 1.00$, 95% CI [-0.05, 0.05], $d = 0.00$), suggesting that participants might be more stable in choice behavior between public and private settings when art-making experts are observing.

For an additional analysis, we combined both conditions and defined the between-group factor audience type and within-group factor public versus private as independent variables. The choice types were then used as dependent variables in two separate analyses. The two-way mixed ANOVA showed that liking choices remained stable with no significant effect of any condition (see S2 Table in Supplementary Information). Wtp showed an interaction $F(1, 121) = 4.95$, $p = .028$, $\eta^2 = .039$ (S3 Table in Supplementary Information), suggesting that being observed by art-making versus art-pricing experts had opposing effects on payment choices. (See also Supplementary Information S3, S4A, and S4B Figs for additional plots showing between subject variability overall separated between art-making and art-pricing groups and for individual variability between the two choice types [for further reading, 94,95].

## Eye-tracking results—Total number of fixations and last fixations as choice predictors

We analyzed two eye-movement measures for our hypothesis 2.a- 2.b: total number of fixations and last fixations. To test our hypotheses 2.c-2.d, we combined the eye-tracking results with the behavioral choice data.

Descriptive statistics, that are all means and standard deviations of total number of fixation and last fixation over all 13 trials between the condition, are reported in S4 Table in the Supplementary Information. The results for the total number of fixations showed that most participants looked mainly to the center of the screen and, thus, to the neutral position. Based on *M*s and *SD*s, the results do not give any indication of differences between the conditions for both the between-group as well as the within-group factors. We therefore could not accept our hypothesis 2.a.

Also, when analyzing change in gaze behavior (hypothesis 2.b) between the within-group conditions (public minus private), we found no significant effect in the total number of

fixations (see Table 2). However, the analysis of the last fixation over the 13 trials indicated an interesting effect in the art-making experts' group. Participants looked more often at high artistic value artworks at the end of the trial (last fixations; Table 2 and Fig 5; for changes in gaze behavior for total number of fixations and last fixation in all conditions see S5 Fig in Supplementary Information) in the public condition. The difference to the private condition was marginally significant ($t$ (60) = 1.98, $p$ = .05, 95% [-0.01, 1.15], $d$ = 0.254).

We tested hypotheses 2.c-2.d, by employing a linear regression to test if the most fixated-on image, and separately the last fixated image, could predict choices on a trial-by-trial basis (separately for liking and wtp). After fitting these four regressions models for each participant, we tested whether the distribution of fitted *beta* parameters was different from zero, indicating a significant relationship between eye-movements and choices. As can be seen in Table 3, all linear regression parameters were significant, supporting our hypothesis 2.c, with a stronger effect in the total number of fixations compared to last fixation.

Interestingly, the last fixation showed no difference between the two choice types ($t$(122) = 0.31, $p$ = 0.761, 95% CI [-0.03, 0.04], $d$ = 0.015). However, for the total number of fixations, there was a difference between liking and wtp ($t$(122) = -2.89, $p$ = 0.005, 95% CI [-0.08, -0.02], $d$ = 0.178), meaning that gaze behavior (number of fixations) predicted liking choices significantly better than wtp choices, supporting our hypothesis 2.d, that participants look most at what they like (being a more personal value decision); whereas they may choose a different artwork considering wtp choices.

Moreover, we analyzed the individually fitted *beta* values from the regression analysis above with a mixed ANOVA to see if there was an effect of audience type (art-making vs. art-

**Table 2. Change of fixations between public and private condition.**

| | | | | 95% CI | | | | |
|---|---|---|---|---|---|---|---|---|
| | **df** | **M** | **SD** | **Lower** | **Upper** | **t** | **p** | **Cohen's d** |
| **Total number of fixations** | | | | | | | | |
| Audience type | | | | high artistic value | | | | |
| Art-pricing experts | 61 | -0.07 | 3.05 | -0.85 | 0.71 | -0.19 | .85 | 0.024 |
| Art-making experts | 60 | -0.64 | 2.81 | -1.36 | 0.09 | -1.75 | .08 | 0.225 |
| | | | | neutral value | | | | |
| Art-pricing experts | 61 | 0.39 | 2.99 | -0.37 | 1.16 | 1.02 | .31 | 0.130 |
| Art-making experts | 60 | -0.13 | 2.06 | -0.66 | 0.40 | -0.49 | .62 | 0.063 |
| | | | | high monetary value | | | | |
| Art-pricing experts | 61 | 0.10 | 2.39 | -0,51 | 0,71 | 0.34 | .74 | 0.043 |
| Art-making experts | 60 | -0.49 | 2.58 | -1.16 | 0.17 | -1.48 | .15 | 0.063 |
| **Last fixation** | | | | | | | | |
| | | | | high artistic value | | | | |
| Art-pricing experts | 61 | -0.19 | 2.22 | -0.76 | 0.37 | -0.68 | .50 | 0.086 |
| Art-making experts | 60 | 0.57 | 2.24 | -0.01 | 1.15 | 1.98 | .05* | 0.254 |
| | | | | neutral value | | | | |
| Art-pricing experts | 61 | 0.03 | 2.32 | -0.56 | 0.63 | 0.11 | .91 | 0.014 |
| Art-making experts | 60 | -2.45 | 2.42 | -0.87 | 0.38 | -0.79 | .43 | 0.189 |
| | | | | high monetary value | | | | |
| Art-pricing experts | 61 | -0.08 | 2.10 | -0.62 | 0.46 | -0.3 | .77 | 0.039 |
| Art-making experts | 60 | -0.23 | 2.51 | -0.88 | 0.42 | -0.71 | .48 | 0.091 |

*Note. M* and *SD* of total number of fixations and last fixations over all 13 trials between the condition
*p < .05 adjusted to multiple comparison.

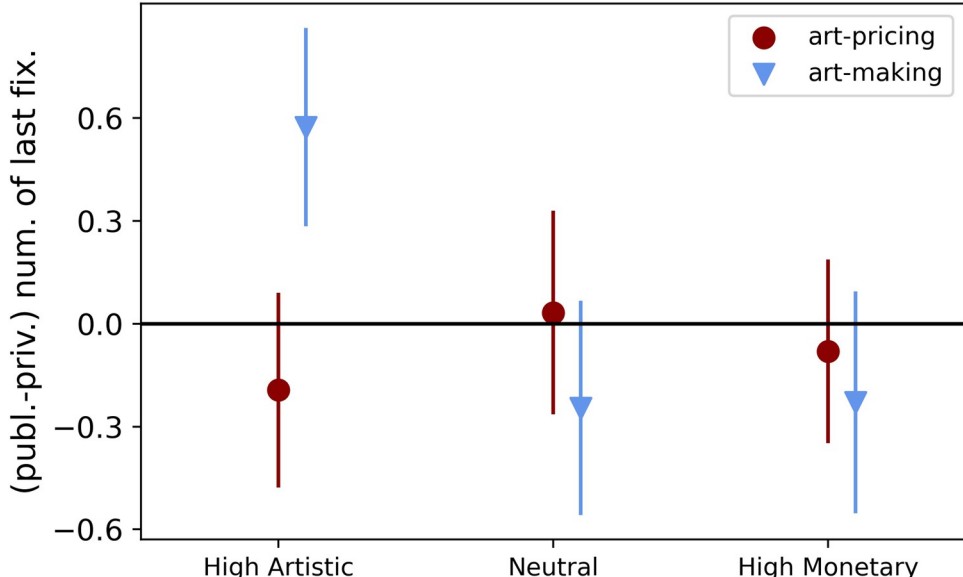

**Fig 5. Change in gaze behavior of the last fixation between the two audience conditions calculated as public minus private.**

pricing) on the relationship between the fixation preferences and choices. Again, the total number of fixations predicted liking choices much better than wtp; there was no effect of audience type, nor an interaction. (Table 4; reported means and standard deviations see S5 Table in Supplementary Information).

## Neuroendocrinological results—Testosterone and cortisol levels driving choice behavior

In an explorative analysis, we analyzed if individual differed in their choice behavior when being observed by experts along hormonal differences indicating social reputational drives and stress. This was measured through levels of testosterone and cortisol. We used the average testosterone from the pre- and post-measurement as the levels did not change significantly and merging both values yielded a less noisy measure for the real baseline testosterone of each person. For cortisol we used the change in cortisol level (i.e., cortisol post experiment measurement minus cortisol sample taken before the experiment). Both represent our neuroendocrinological factors.

A first analysis focused on the relationship between testosterone and cortisol change. A Pearson correlation ($r = -0.46$, $p = < .001$; Spearman correlation $r = -0.4$, $p = < .001$) showed

**Table 3. Simple linear regression model for predicting liking and willingness-to-pay (wtp) choices.**

| Choice types | df | M (b) | SD (b) | 95% CI Lower | Upper | t | p | Cohen's d |
|---|---|---|---|---|---|---|---|---|
| | | | | Total number of fixations | | | | |
| Liking | 122 | 0.505 | 0.259 | 0.43 | 0.53 | 18.96 | < **.001** | 1.710 |
| Wtp | 122 | 0.455 | 0.256 | 0.39 | 0.48 | 17.91 | < **.001** | 1.615 |
| | | | | Last fixation | | | | |
| liking | 122 | 0.073 | 0.323 | 0.02 | 0.13 | 2.50 | **.014** | 0.225 |
| Wtp | 122 | 0.068 | 0.324 | 0.01 | 0.13 | 2.33 | **.022** | 0.230 |

**Table 4. Mixed ANOVA using total amount of fixation (image participants mostly looked at) showing differences between audience type and choice type.**

| Variables | F (1,121) | p | η² |
|---|---|---|---|
| Audience type (art-making/art-pricing experts) | 0.84 | .36 | 0.007 |
| Choice type (liking/wtp) | 8.26 | < .01 | 0.064 |
| Interaction | -0.01 | 1.00 | 0.000 |

that higher testosterone was negatively correlated with cortisol change, meaning higher testosterone predicted larger cortisol decrease. This supports the dual-hormone hypothesis (e.g., [75–77]). It also warrants that we proceeded with our analysis.

Based on the dual hormone hypothesis [75–77], we expected that participants with relatively high testosterone, but decreased cortisol levels behave differently. To this end, we used the mean testosterone levels and the change (usually decrease, see above) in cortisol levels, to cluster participants into two groups. Clustering was performed with a Gaussian mixture model (2 components, 10 initializations, full covariance), using the *GaussianMixture* function of the *scikit-learn* library [78]. This clustering resulted in two groups with distinct hormonal profiles (Fig 6): Cluster 1 (*n* = 30) had a mean testosterone levels of 78.91 +/- 21.69 *pg/ml* and a cortisol change of -0.36 +/- 0.64 *nmol/l*. Quite different, Cluster 2 (*n* = 20) had a mean testosterone level of 140.01 +/- 46.59 *pg/ml* and a cortisol change of -3.53 +/- 2.55 *nmol/l*. Descriptive results are reported in Table 5 and presented in Fig 6.

Next, this grouping was used to visualize the differences in choice making strategies for the within subject condition (Fig 7A). According to the descriptive statistics, Cluster 1 participants choose mostly high artistic value artworks for both choice types. Direction of value choice remained similar in the public condition. Choices made by participants belonging to Cluster 2 (participants with high testosterone/high cortisol decrease levels) averaged their choices mostly around the middle artwork in the private condition. However, in the public condition, the two choice types diverged with remaining liking choices for artworks higher of artistic value between private and public; though participants were more wtp for high monetary artworks in the public condition compared to the private setting.

Fig 7B presents the combination of both, the within- and the between-subject conditions. The results show that Cluster 1, while being watched by art-pricing experts, had similar liking choices compared to the private condition. In addition, wtp choices changed from high artistic to high monetary value artworks, while being watched by the art-pricing audience. Interestingly, Cluster 1 showed also stable liking choices also in the art-making expert condition. However, these participants were more wtp for high artistic value artworks, while being watched by the art-making experts, where in the private condition the choices were averaged in the middle.

Cluster 2 showed a potential influence of socio-cultural trained behavior (as discussed in the Introduction; see also Fig 2), especially in the art pricing expert group by choosing in the private condition along the following pattern: liking more high artistic value artworks and wtp for high monetary artworks. This result was more pronounced in the public condition, where both choice types diverged even stronger. Latter would suggest the first strategy we discussed in the Introduction. However, within the art-making audience group all choices were biased towards high monetary value artworks, potentially following the second strategy and not complying with the audience believed value system. Since this division was based on a small sample size (*n* = 50) considering neuroendocrinological measures, we do not compare the groups with inferential statistics, but only provide the descriptive statistics in Table 5.

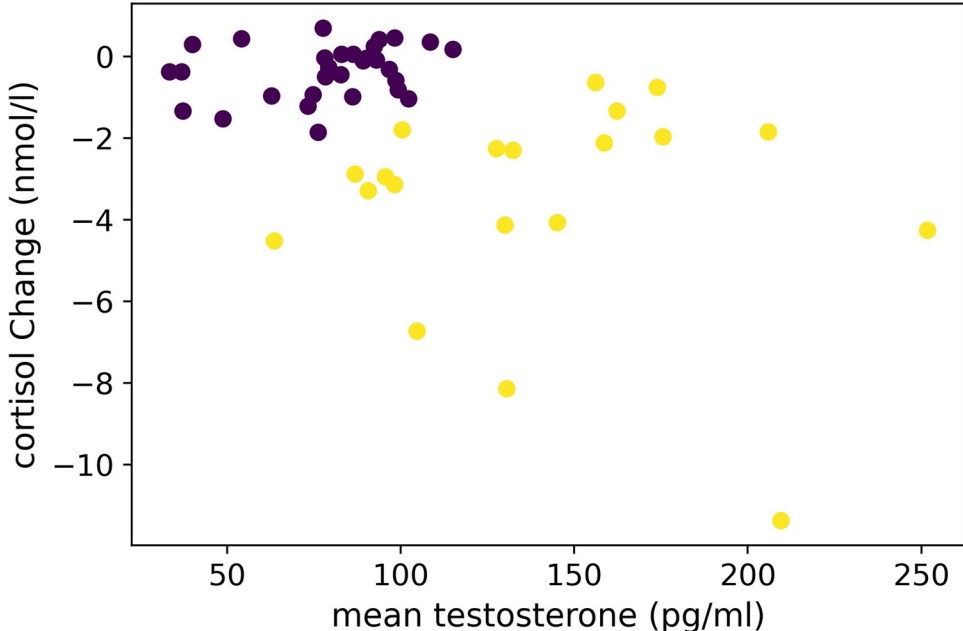

**Fig 6. Clustering of groups respecting both association of testosterone and cortisol interrelations.**

## Results of motivational factors—Behavioral activation and anxiety scale as choice predictors

In the subsequent analysis, we used the BIS Scale as well as the summed score of the BAS Drive and BAS Reward Response subscales. Latter BAS subscale has been reported as measure of trait dominance [79] and is in accordance with previous studies suggesting an association with testosterone effects under stress influence [60,61] and respecting the dual-hormone-hypothesis [73,74]. Furthermore, the LSAS anxiety scale [83–85] was exploratively included. We conducted the following three analyses and included for all three analyses BIS, BAS (Drive + Fun), LSAS-anxiety, and between-group condition as predictors: (1) a linear regression analysis to predict the difference between liking and wtp choices. There was a significant effect for the BAS (Drive+Fun) Scale ($B = 0.07$, $SE = 0.02$, 95% CI [0.02,0.12], $r2 = 0.08$, $t = 2.78$, $p<0.01$)

**Table 5. Descriptive analysis means and standard deviations of liking and willingness-to-pay (wtp) choices for all conditions for both hormone clusters.**

| Choices | Cluster groups | Cluster 1 | | Cluster 2 | |
|---|---|---|---|---|---|
| | within-/between-group factors | *M* | *SD* | *M* | *SD* |
| | Private | | | | |
| liking | Art-pricing experts | 1.93 | 0.5 | 1.93 | 0.18 |
| | Art-making experts | 1.88 | 0.19 | 2.01 | 0.16 |
| wtp | Art-pricing experts | 1.91 | 0.29 | 2.03 | 0.30 |
| | Art-making experts | 1.98 | 0.22 | 2.03 | 0.18 |
| | Public | | | | |
| liking | Art-pricing experts | 1.90 | 0.24 | 1.87 | 0.18 |
| | Art-making experts | 1.88 | 0.20 | 2.02 | 0.15 |
| wtp | Art-pricing experts | 2.05 | 0.30 | 2.12 | 0.28 |
| | Art-making experts | 1.89 | 0.28 | 2.10 | 0.13 |

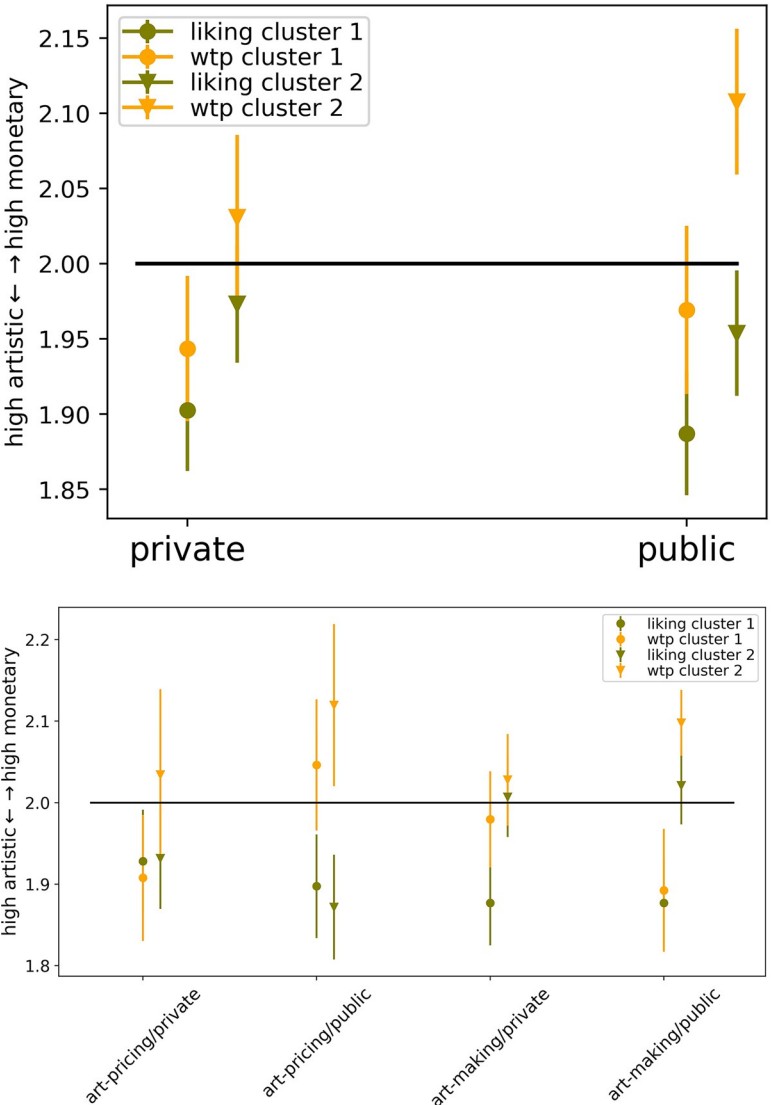

**Fig 7. Descriptive statistics of choice behavior along with both Clusters.** (A) Choices between the within-subject condition private vs. public. (B) Within and between-group choice behavior; for visibility, Cluster 2 is shifted slightly to the right along the x-axis for each condition.

(for full results see S6 Table). (2) a linear regression analysis to study the change in liking choices between public versus private decisions. None of the predictors were significant (see Table 6). (3) a linear regression analysis to study the change in wtp choices between public vs. private decisions (see Table 6). Again, the BAS (Drive+Fun) Scale ($B = 0.06$, $SE = 0.03$, 95% CI [0.01,0.11], $r2 = 0.10$, $t = 2.19$, $p < 0.03$) could predict wtp choices and there was also a significant relation with the between-group factor as reported before.

## Discussion

By implementing a novel paradigm, we studied how liking and willingness-to-pay (wtp) choices differed when made within differing social contexts involving a private and a public

**Table 6. Linear regression model for predicting liking and willingness-to-pay (wtp) choices.**

| Liking | B | SE | 95% CI | $r^2$ | t | p |
|---|---|---|---|---|---|---|
| Intercept | -0.15 | 0.14 | [-0.43,0.13] | 0.01 | -1.05 | 0.30 |
| BIS | 0.03 | 0.04 | [-0.48,0.10] | 0.01 | 0.74 | 0.46 |
| BAS (Drive + Fun) | 0.01 | 0.02 | [-0.02,0.05] | 0.01 | 0.70 | 0.49 |
| LSAS Anxiety | -0.00 | 0.00 | [-0.00,0.00] | 0.01 | -0.41 | 0.68 |
| art-making/art-pricing experts | -0.00 | 0.03 | [-0.07,0.05] | 0.01 | -0.27 | 0.79 |
| Wtp | B | SE | 95% CI | $r^2$ | t | p |
| Intercept | -4.08 | 0.19 | [-0.78,-0.03] | 0.10 | -2.16 | **0.03** |
| BIS | 0.03 | 0.05 | [-0.07,0.13] | 0.10 | 0.67 | 0.51 |
| BAS (Drive + Fun) | 0.06 | 0.03 | [0.01,0.11] | 0.10 | 2.19 | **0.03** |
| LSAS Anxiety | 0.00 | 0.00 | [-0.00,0.00] | 0.10 | 0.21 | 0.83 |
| art-making/art-pricing experts | -0.10 | 0.03 | [-0.17,-0.02] | 0.10 | -2.42 | **0.01** |

*Note.* Included predictors were BIS, BAS (Drive + Fun), LSAS-anxiety, and between-group condition.

setting—the latter suggesting to participants that they were being observed by one of two distinct art reputation audience groups (art-making or art-pricing experts).

## Behavioral results

As expected, overall, participants liked high-artistic value artworks more and were more often wtp for high monetary artworks. The results are in accordance with our denotation made in the Introduction (see also Fig 2 Stage I-II), that such art choices are guided by socio-cultural learned associations (see [8,12–16,30], see also [96]). Interestingly, liking choices were less influenced than wtp choices in all conditions (within-group factor private vs. public; between-group art-making/art-pricing audience) leading to the result (hypothesis 1.a) that there was no detectable change in choice behavior when the art-making experts were watching. Hence, liking appears to be a more stable—and most likely due to its personal connotation—individual-centered parameter of choice preferences, supporting many studies made in art research [7,8, see for review 10, see also 29,43–48]. The influence of art-pricing experts and monetary issues, however, triggered participants to choose more often art high in monetary value in the public context (hypothesis 1.b), following our suggestion that monetary aspects may have a more direct connection to social reputation factors. We would also note that participants followed the direction of our suggested socio-culturally learned behavior ([11–22,37], learned associations with values and audience representatives) and that our study population (artistical laymen) were prone to the opinions or believed value-system of the experts, as past studies reported [9,16,25,30,46–48]. This is also visible in the slight aversion to the neutral position (S1A and S1B Fig in Supplementary Information). This, however, should be further investigated and leads to one interesting implication, also regarding the social reputation framework, to test this study also with art experts compared to laymen.

Hypothesis 1.c. was also supported, showing that choices remained unchanged if the audience did not match the choice variable (or believed value system). This means if the audience were art-pricing experts liking did not change from private to public, and the wtp remained stable regardless of the existence of art-making experts. We therefore accept our main hypotheses that social reputation had an influence on both choice types according to associated audience and values, where the values had a stronger influence than the audience. However, also here we found in choice behavior considering monetary aspects changes due to audience influence.

## Eye-tracking results

To summarize our general, choice-independent eye-movement analyses: we did not find increased fixations in the public condition compared to private, thus hypothesis 2.a was not confirmed. Regarding change in fixations between the public versus private conditions, we only found a marginally significant effect with last-fixations focusing on more artistic value artworks within the art-making audience group. Even though it was only a marginally significant effect, it is interesting for future studies that artistic observers motivate people to especially look in the end to high artistic value art, considering a potential impact of context influences to initiate further in-depth evaluation of the artwork [e.g., 13–17,37].

One key potential issue here is that participants appeared to mainly look at the center. This was presumably due to our study design, where first the fixation cross was presented in the center and then the middle image was seen at the same position after the fixation cross (landing position, see for further reading [80,97–100]). Likewise, the images of the artworks were oriented horizontally, to the extent that many fixations were probably performed over the center to one side or the other. However, since our study was particularly focused on fixation behavior in terms of quantity of fixations in general, and especially in terms of predicting choice behavior, the effect of position in terms of eye movements was not too relevant to the results. The effect of position with respect to the made choices is therefore unlikely because both paintings were randomly positioned between participants and the participants selected the artworks in a relatively evenly distributed manner (equal probability with respect to position, see S1A and S1B Fig in Supplementary Information).

In a next step, we focused our analysis on how fixations relate to artwork choices (see [62–67]). Furthermore, the temporal specification was interesting, because, besides the total number of fixations as a predictor for choice behavior, we expected fixations to shift towards especially the most liked artworks at the end of the presentation period [66–67, see also 80]. Our results showed that the total number of fixations per trial predicted choice behavior (hypothesis 2.c). That is, the artwork that they looked at most often was chosen in the end. This result was highly significant and delivers a bases for future research to use this implicit measure as indicator for choice-behavior. We also found that the last fixation predicted choice behavior. However, this result was not as pronounced as the total number of fixations.

Regarding differences in liking and wtp choices and fixations (hypothesis 2.d), we tested the assumption that personal liking would be more strongly associated with gaze behavior than wtp choices. We did find this general pattern: eye movement fixations had a significantly stronger relationship with liking compared to wtp choices, especially in the public-condition. Interestingly, the effect seemed to be stronger when one assumed to be observed by art-pricing experts; however, the difference between the two audience was not significant. This result is of particular interest for future studies investigating eye-movements for the two choice types separately (which we could not dissociate in our study as choices were taken consecutively, see limitations below).

## Neuroendocrinological results

The explorative results for the hormonal analysis revealed the following overall results: participants from Cluster 1 (average testosterone/cortisol change) showed in general converging choice behavior, except when art-pricing experts were observing in the public condition. Cluster 2 (participants with high testosterone/high cortisol decrease levels), on the contrary, appeared to follow socio-cultural learned behavior ([8,10–17,22–24], see also Fig 2), especially when art pricing experts were assumed to be watching, with diverging results of the two choice types, which was strengthened in the public condition (see also for public choice behavior,

[30,52,58,59]). In the art-making audience group, however, choice behavior appeared to converge in general, with an overall tendency to choose more high monetary value artworks.

Overall, we could observe that neuroendocrinological data was more associated with monetary issues respecting both monetary value and art pricing experts. This raises some interesting implications, namely that testosterone/cortisol—as shown in previous studies addressing economic issues (e.g., [58–60,70,71])—are susceptible to ownership and the monetary reputational potential of art. Choices of liking or personal appreciation, however, might be related to possibly other hormonal or neurotransmitter functioning, where further research is necessary to investigate this aspect (see for discussion [82]).

## Motivational factors results

We included the scales measuring motivational factors as proxy measures for the neuroendocrinological measures, as the above did not include the female participants resulting in a rather small sample. Especially relevant were the BAS-scales [60,61,79] results, which showed significant results in differences in liking and wtp choices (see also [65,78,83]), and thus seems promising for future research. Interestingly, liking choices could not be predicted by this scale, which hints again to the idea that personal liking appears to be differently influenced than monetary issues—or wtp—depending on the given context [8–10,18–26,30,31,46–48]. Hence, we found a positive association with the BAS scale, which could explain 10% variance of the wtp choices. This aligns with former research showing that monetary values are strongly triggered by reputational status-seeking behavior [58–61,70,71]. We further did not find any differences between the private versus public condition, suggesting that high reputation seeking participants might show similar personality-trait influence in both conditions.

Last, we did not find any association with both the behavioral inhibition scale [79] and the LSAS [83]. We can only assume that anxiety or inhibition behavior might lead to a translated form of flight or fight behavior, where choice behavior becomes rather irrational or that in some cases participants confirmed, in some cases they took an opposing position, explaining the found non-association. However, interpretation remains difficult. Further in-depth research, also in combination with stress-levels, would be needed. Latter would also go in line with former research associating the LSAS with cortisol response during social avoidance behavior [84,85].

## Limitations, emphasis, and final conclusion

Unquestionably, new study designs come also with caveats and often new questions for future research. One major limitation of the study-design, for the behavioral results, was that the two dependent variables—liking and wtp—were asked consecutively, potentially causing some interdependencies. For a follow-up replication study, the choices could be asked separately, maybe even at two different time points. This issue may explain the small effect size of the main effect between the two choice types, liking and wtp. As we asked the two choice types consecutively after the viewing session (and the eye-tracking recording) we were also not able to analyze the eye-tracking patterns separately for the two choice types. Again, separating the trials could systematically disentangle eye-movements between the two choice types, which should be adapted in future, but also in replication studies. Furthermore, we cannot be sure that all subjects accepted the definition of artistic quality that was specified in the instructions (see description S1 File for German and S2 File for English in the Supplementary Information). In contrast to monetary values, the definition of artistic value has always held a certain individual freedom in regard to meaning [38–40,94,101]. Nevertheless, we tried to reduce potential

confounding as much as possible through the descriptions, and participants did not express any confusion about the meaning of artistic value in this study context.

Considering stimuli level, despite the careful selection of stimulus triplets, we cannot exclude that low-level visual features in the artworks may have partially influenced choice behavior. However, based on the study by Massaro and colleagues [87, see also 88], this study also discussed the stronger top-down influence of social-cultural learning on decision-making and gaze behavior compared to bottom-up driven low-level feature influence. Nonetheless, further studies dissociating the aspects between weight on rather bottom-up objective features and top-down social-cultural influences would be an exciting direction of study [see also 88,102–110].

The time intervals of the neuroendocrinological measures between pre- and post-measurements were sometimes relatively long (depending on how much time participants took to make their choices). Thus, the stress levels of some of the participants could have decreased already due to natural hormone-level regulation [58,73]. Future studies might focus especially on the temporal issues involved. This also leads to some further implications for our study design: although nearly all participants stated after the study during the clarification that they were believing the cover story and life camera, still, our design could benefit from a more stressful public context (e.g., real in-person presence of the observers).

However, as our results showed very promising implications using neuroendocrinological measures for art research, we suggest dedicating own studies to this intriguing measure also in the field of art research with a larger number of subjects and sampling. For example, as we could find that social reputation influenced choice behavior, future intervention studies with testosterone administration—triggering heightened reputation-seeking behavior—could investigate more precisely whether choice behavior changes only along monetary issues or with wtp choices; but liking, as factor of personal appreciation, might be less influenced by the social context and relations. However, this leaves an open question to what neuroendocrinological measure could potentially be the driving factor for liking and artistic preferences (e.g., a promising candidate could be dopamine; see for discussion [81,82]).

Furthermore, for art and aesthetic research as well as social psychological studies, using prestigious objects as stimuli, investigations should focus more on social mechanisms and how they lead to behavioral change. Hence, we would like to emphasize that interactions between the private self and self-image, and within a particular social context (social self), are influenced by social reputation, beyond other social factors, and in consequence can modify choices, judgments, and ratings for all kinds of precious cultural goods (see [2–5,10,34]); precious cultural goods that are valued most likely due to human's intrinsic sense of aesthetic sensitivity [38–40]. In this sense addressing artworks with values coordinating our choices within private and public contexts might intuitively appear not surprising, though has never been studied empirically before considering the entangled values surrounding aesthetic commodities. Herein, artworks appear to have an interesting position due to their cultural heritage as purchasable goods but also as 'only hangings' in museums; much different then aesthetic designs products like for example stylish cars [88]. Hence, with artworks, choices such as liking and willingness-to-pay as wish for ownership conceal other complex patterns of interrelationships between values, social reputational frame, and the two choice types, which, as Berridge and colleagues [23,24] noted, require much further investigation. Our study presents one of the first empirical steps for such investigations and hereby also merging social economic and art research aspects.

That said, via a newly developed paradigm in art research, our study demonstrated the advantages of implicit measurements and multi-method procedures in art research considering aspects of social influence. The combination of the measurements reveals a deeper

understanding and a more precise interpretation of cognitive and affective processes, which are often not visible in studies measuring valuations only. Furthermore, we were able to gain insights into art evaluation processes suggesting that interactions with art objects are a reciprocal dynamical process, where choices represent a complex interplay between the specific environment, its socio-cultural value systems, and the self-selecting person.

## Supporting information

**S1 Fig. A. Descriptive analysis of position and value position.** The dashed horizontal lines show chance ($1/3 \approx 33.3\%$) choice. Error-bars represent 2 standard errors of the mean. A. Effect shows the average choice percentage for the three locations (x-axis) separated by choice type (see legend). **B. Descriptive analysis of position and value position.** The dashed horizontal lines show chance ($1/3 \approx 33.3\%$) choice. Error-bars represent 2 standard errors of the mean. B Effect shows the average choice percentage for the three stimulus types (x-axis) separated by choice type (see legend).
(TIF)

**S2 Fig. Descriptive analysis liking and willingness-to-pay choices for all conditions.**
(TIF)

**S3 Fig. Between subject variability overall separated between art-making and art-pricing expert groups.** Each dot represents the average preference for stimulus type (y-axis) for a participant for liking and willingness-to-pay choices (x-axis).
(TIF)

**S4 Fig. A. Between subject variability reported for both choice types.** Gray lines connect dots within participants. The dashed line shows the—on average—neutral choice. A. Art-making group. **B. Between subject variability reported for both choice types.** Gray lines connect dots within participants. The dashed line shows the—on average—neutral choice. B. Art-pricing group.
(TIF)

**S5 Fig. Eye-tracking results.** Left: Total number of fixations and last fixations in the different conditions. Right: Change in gaze behavior between the two audience conditions calculated as public minus private. Upper figures show results for total number of fixations. Figures below for last fixations.
(TIF)

**S1 Table. List of used artworks and assignment of the artworks in the sets.**
(PDF)

**S2 Table. Two-way mixed ANOVA for liking.** Independent variable between-participant factor audience type and within-participant factor public vs. private. Dependent variable choice type liking.
(PDF)

**S3 Table. Two-way mixed ANOVA for liking.** Independent variable between-participant factor audience type and within-participant factor public vs. private. Dependent variable choice type willingness-to-pay.
(PDF)

**S4 Table. Mean and standard deviations of total number of fixation and last fixation over all 13 trials between the condition.**
(PDF)

**S5 Table. Mean of beta and standard deviations of total amount of fixations between the two audience type conditions and for both choice types, liking and willingness-to-pay (wtp).**
(PDF)

**S6 Table. Linear regression model for predicting liking versus willingness-to-pay (wtp) choices.** Included predictors were BIS, BAS (Drive + Fun), LSAS-anxiety, and between-participant condition.
(PDF)

**S1 File. Cover Story (in German).**
(PDF)

**S2 File. Cover Story (in English).**
(PDF)

## Acknowledgments

The plan for this study and its design was a joint project between Blanca T.M. Spee, Helmut Leder, and Christoph Eisenegger, who very sadly and totally unexpectedly passed away in 2017.

## Author Contributions

**Conceptualization:** Blanca T. M. Spee, Christoph Eisenegger, Helmut Leder.

**Data curation:** Blanca T. M. Spee, Jozsef Arato, Jan Mikuni, Ulrich S. Tran.

**Formal analysis:** Blanca T. M. Spee, Jozsef Arato, Ulrich S. Tran.

**Investigation:** Blanca T. M. Spee.

**Methodology:** Blanca T. M. Spee, Christoph Eisenegger, Helmut Leder.

**Project administration:** Blanca T. M. Spee.

**Software:** Jozsef Arato.

**Supervision:** Matthew Pelowski, Helmut Leder.

**Validation:** Blanca T. M. Spee, Matthew Pelowski, Ulrich S. Tran, Helmut Leder.

**Visualization:** Blanca T. M. Spee.

**Writing – original draft:** Blanca T. M. Spee, Helmut Leder.

**Writing – review & editing:** Blanca T. M. Spee, Matthew Pelowski, Jozsef Arato, Jan Mikuni, Ulrich S. Tran, Helmut Leder.

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
