## [Decision Letter · Decision Letter 0]

22 Nov 2021

PONE-D-21-26393

Social Reputation Influences on Liking and Willingness-to-Pay for Artworks: A Multimethod Design Using Behavioral, Physiological, and Psychometric Measures

PLOS ONE

Dear Dr. Spee,

Thank you for submitting your manuscript to PLOS ONE. After careful consideration, we feel that it has merit but does not fully meet PLOS ONE’s publication criteria as it currently stands. Therefore, we invite you to submit a revised version of the manuscript that addresses the points raised during the review process.

The two reviewers coincide in that the manuscript has merits, the research idea is sound, and the text is well written. However, they also describe several concerns and suggestions to improve the manuscript. These comments should be addressed before the manuscript is ready for publication. Let me highlight the most relevant elements from the reviews.

First, R1 offers an alternative explanation for the results: can the data be explained by a preference for the middle-position? This is an important issue that must be discussed and addressed. If this possibility was not taken into account, it will posit serious limitations to the conclusions.

Both reviewers coincide on asking for more information about several aspects of the study: R1 needs a better justification for some of the variables/measures, and for the use of cluster analyses techniques, while R2 suggests a more comprehensive literature review and gives some references. I think these suggestions are highly pertinent. Thus, I would encourage you to rewrite the Introduction section to include some additional references, and to revise the rest of the sections to provide a good justification for the measures and analyses. It would be advisable to try to highlight those aspects of the research that are new or original (as R1 comments, some of the results seem trivial/obvious at first glance).

I appreciate the inclusion of sensitivity analyses. However, as R2 indicates, it would be good to have a description of how the sample size was chosen (constraints, goals...).

Decision: I am rejecting the manuscript and inviting to resubmit a revised version that addresses the concerns raised by the two reviewers.

We look forward to receiving your revised manuscript.

Kind regards,

Fernando Blanco

Academic Editor

PLOS ONE

3. Please amend the manuscript submission data (via Edit Submission) to include author Christoph Eisenegger.

Reviewers' comments:

Reviewer's Responses to Questions

**Comments to the Author**

1. Is the manuscript technically sound, and do the data support the conclusions?

Reviewer #1: Partly

Reviewer #2: Yes

2. Has the statistical analysis been performed appropriately and rigorously? 

Reviewer #1: Yes

Reviewer #2: No

3. Have the authors made all data underlying the findings in their manuscript fully available?

Reviewer #1: No

Reviewer #2: No

4. Is the manuscript presented in an intelligible fashion and written in standard English?

Reviewer #1: Yes

Reviewer #2: Yes

5. Review Comments to the Author

Reviewer #1: Social Reputation Influences on Liking and Willingness-to-Pay for Artworks: A Multimethod Design Using Behavioral, Physiological, and Psychometric Measures

The research reports a novel study examining the willingness-to-pay and liking for artworks, in private and public conditions. Public choices were made in the context of either an art expert or pricing expert and the artworks had an aesthetic and pricing value specified by the relevant experts.

The findings show that public willingness-to-pay choices were more influenced by the presumed presence of a pricing expert, whereas liking choices were not, appearing to be more of a personal decision. In addition, willingness-to-pay was influenced by the apparent value of the artwork and liking was also influenced by their apparent aesthetic quality.

While the work has a number of merits it also has some issues that prevent me from recommending publication in its current form. Some of the findings do not seem to contribute greatly to the field. The finding that people are willing to pay more for (apparently) more valuable artworks is not particularly surprising. Similarly, the influence of expert opinions of the aesthetic liking of artworks is not a novel finding (see Kirk et al., 2009, as cited by the authors in the introduction). However, the use of hormonal data certainly adds an interesting element to the research. However, the rationale for the inclusion of the ‘Behavioral Activation and Anxiety Scale as Choice Predictors’ needed to be stronger. The ‘explorative inclusion’ of the LSAS anxiety scale appears overly speculative and it wasn’t clear what this added to the study.

A major concern is the design and task that was used. An item’s position has an influence on whether it will be chosen, with there being a bias/preference to choose middle options (Valenzuela & Raghubir, 2009; Atalay et al. 2012) and it is often important to control for effects of position when a choice is made from similar options. Unfortunately in this design the effect of position was not controlled for because the middle artwork always had an artistic value and price that was in the middle, and so it is not possible to disambiguate the effect of position from the effect of the artwork’s attributes (artistic/price) on the liking/willingness-to-pay choices. The Means from table 1 are all close to the neutral value (2). Does this indicate frequent choice of the middle artwork? As the authors note, these means show that there was not much effect of condition. In relation to the effect of position how often was each artwork (in each position) chosen? This information does not seem to be included in the tables provided.

It is also not clear from the manuscript why a middle option was used in the design. It is not clear from the description what it added and how it enabled a comparison of the key variables of monetary value/aesthetic quality, when the left/right options already did this.

In addition, there is substantial evidence that participants have a bias to look at the middle item (e.g. Tatler, 2007) and the eye tracking results show this expected middle looking bias in terms of more looks and last looks. The literature on the middle position and choice, and gaze behaviour, is not referred to and should be included (e.g. Atalay et al., 2012; van der Laan, et al., 2015). The possible impact of these effects on the results should also be considered in detail. In particular, it is possible that the gaze results predicted choice because the participants were looking more at the middle artwork. That is, was it an effect of position rather than artwork?

The width of the artworks varied (height was kept constant), how were the authors sure that the participants were looking at a particular artwork, when the width varied from one trial to the next? How was the region of interest controlled?

Why was a cluster analysis used for the hormonal analysis, rather than a regression? What was the rationale of creating two groups rather than treating hormone levels as a continuous variable?

Participants were asked to make two choices on a trial: liking/willingness-to-pay. It is not clear from the information provided whether participants could choose the same artwork for both choices or had to choose a different artwork. The authors should include the task instructions for clarity.

Fig S2 in supplementary Materials – did not seem to be included

Table 2 – Eye tracking analysis. Were these analyses corrected for multiple comparisons?

Line 154 – ‘group group’

References:

Atalay, A., Bodur, H., & Rasolofoarison, D. (2012). Shining in the center: Central gaze cascade effect on product choice. Journal of Consumer Research, 39(4), 848–866.https://doi.org/10.1086/665984

Kirk U, Skov M, Hulme O, Christensen MS, Zeki S. Modulation of aesthetic value by semantic context: An fMRI study. Neuroimage. 2009; 44(3): 1125-1132. doi: 10.1016/j.neuroimage.2008.10.009

Tatler, B. W. (2007). The central fixation bias in scene viewing: Selecting an optimal viewing position independently of motor biases and image feature distributions. Journal of Vision, 7(14), 4. https://doi.org/10.1167/7.14.4

Valenzuela, A., & Raghubir, P. (2009). Position-based beliefs: The center-stage effect. Journal of Consumer Psychology, 19(2), 185–196. https://doi.org/10.1016/j.jcps.2009.02.011

van der Laan, L., Hooge, I. T., de Ridder, D. T., Viergever, M., & Smeets, P. A. (2015). Do you like what you see? The role of first fixation and total fixation duration in consumer choice. Food Quality and Preference, 39, 46–55. https://doi.org/10.1016/j.foodqual.2014.06.015.

Reviewer #2: The MS states in line 307 "Whereas representational paintings might be expected to have a higher, positive correlation between price and artistic quality". There is a debate about 'what' is artistic quality. As artistic quality is not a quality of the object but an individual assessment. I suggest to reframe the as perceived or assessed artistic quality.

In line 58, authors discuss about "scholarly discussions of judgment and taste". In my opinion, MS would improve with the addition of modern empirically based discussion of judgement and taste. There were even updates and criticisms of the Eysenck model of judgement cited in 19 (see 10.1016/j.paid.2017.05.041 , https://doi.org/10.1111/bjop.12427, https://doi.org/10.1111/bjop.12440, 10.1093/oxfordhb/9780198824350.013.40). Also, would be interesting to cite more recent views of neuroaesthetics and empirical aesthetics references (see https://doi.org/10.1016/j.cub.2018.06.004 and https://doi.org/10.1016/j.tics.2014.03.003)

In my opinion, the use of "personality" to refer to BIS and BAS could be misleading. Maybe, changing to "personality" to "individual differences" or traits could improve the clarity of the study.

Also, in line with my previous comment, I think that referring to validated scales as "psychometric" is also misleading. Psychometrics usually refers to scientific study of psychological constructs measurement and assessment, while it is related to scale development, is not constrained to it.

Authors state (line 394) that "Despite the discrete (1-2-3) choice options, once averaged across trials, the responses were normally distributed. This was confirmed by a Shapiro-Wills test, which found no deviation from normality (liking: W = .985, p = .197, wtp: W = .991, p = .591)". In general, I think that the analysis would benefit from linear mixed modelling as variation from participants and stimuli could be modelled (see 10.2190/6780-361T-3J83-04L1). Mixed models are specially suitable for empirical aesthetics research as stimuli (usually artworks) are a complex source of variability which not taken into account when using ANOVAs and t.test.

Also, I would suggest to update the current plots to a more informative ones with, at least, participant level information instead of point estimates with CI (see https://doi.org/10.1371/journal.pbio.1002128)

I value the effort of authors to be clear and honest about the exploratory stage of the study presented, the sensitivity analysis and the no a priori power determination. However I think that including the practical sample size determination (was budget?) would complete the "Participants" section.

Regarding the eye tracking results, authors interpretation of data is "Studies have resoundingly shown that individuals spend more time looking at artworks which they find aesthetically appealing and which they liked most (in term of total fixation, see, e.g., [49, 51-54]; or along longer fixation, see [50]). Moreover, participants appear to fixate the preferred image when reaching the decision-moment [53-54]." . But, there are other factors like complexity, interpretability, etc of the stimuli which could play a role (see 10.1177/0301006615596882, 10.1371/journal.pone.0037285). Maybe, taking into account other explanations could improve the MS.

Do authors ensured that participants were naïve regarding art knowledge? Were any measure of it taken?

Note: I don't feel validated to assess the procedure, methods and results from the endocrinological part of the study.

Guido Corradi

6. PLOS authors have the option to publish the peer review history of their article (what does this mean?). If published, this will include your full peer review and any attached files.

Reviewer #1: No

Reviewer #2: **Yes: **Guido Corradi

---

## [Author Response · Author response to Decision Letter 0]

22 Jan 2022

We would like to thank the two reviewers and the editor for their careful review of our paper and for the very useful suggestions. We are happy that the paper was largely well-received in the first iteration and hope that, based on the suggested changes and expansions, it is now even better. 

Below, please find our replies to the individual comments and suggestions, using the following procedure: We first list the comments (in verbatim, shown in italics) from the reviewers, divided up here-and-there by ourselves for conceptual consistency; this is followed by our reply with specific page numbers for corresponding in-text changes in the revised manuscript. In addition, we have also highlighted major changes in the manuscript itself in red text. We have also given the paper an overall edit for language and typos.

Thank you again. We look forward to further working with you all towards a successful publication!

The authors

Reviewer #1: 

The research reports a novel study examining the willingness-to-pay and liking for artworks, in private and public conditions. Public choices were made in the context of either an art expert or pricing expert and the artworks had an aesthetic and pricing value specified by the relevant experts.

The findings show that public willingness-to-pay choices were more influenced by the presumed presence of a pricing expert, whereas liking choices were not, appearing to be more of a personal decision. In addition, willingness-to-pay was influenced by the apparent value of the artwork and liking was also influenced by their apparent aesthetic quality.

While the work has a number of merits it also has some issues that prevent me from recommending publication in its current form. Some of the findings do not seem to contribute greatly to the field. The finding that people are willing to pay more for (apparently) more valuable artworks is not particularly surprising. 

Reply: We would like to thank the reviewer for the kind words and helpful suggestions. We agree that the results of the choice behavior (i.e., liking high artistic value/being willing to pay for high monetary value art) might appear somewhat obvious. However, we would also argue that this is a topic in need of actual empirical research, both in general, and especially in combination and as these interact with social influence.

It is true that a good deal of research has been done with liking and contextual aspects of art—for example, relationship with artistic style (Belke & Leder, 2006, Leder & Nadal, 2014), familiarity of the creator (Kirk et al., 2009), etcetera. However, willingness-to-pay for art has almost never been investigated in the field of art research (see also Lauring et al., 2016). Nor has liking been considered in conjunction with purchase willingness. This is despite persistent arguments that these aspects might play different roles (i.e., liking and wanting, etc.). In addition, and also surprisingly to us after a literature review, while the aspect of social reputation has been addressed in a more general (Beckert & Rössel, 2013; Detotto et al., 2020; Harrington, 2004; Throsby, 1994) or in a philosophical context (Benjamin 1996/1935; Bourdieu, 1968/1934), its influence on the two choices types has not been addressed. The differences or similarities both types of choices carry have been highlighted by Berridge and colleagues (1998; 2009). However, a dissection of the “components of reward” (how Berridge refers to them) has not yet been considered within an experimental design; nor has a study shown clearly that due to contextual reasons, i.e. social reputation the two choice types can either appear as synonymous or contradict. 

Hence, our study is the first to deliver interesting implication of the influence of social reputation on choice behavior, which was not yet empirically tested. In addition, while other aesthetic objects (especially in the consumer world of technical goods) with beautiful design (cars, computers, technical design, etc.) have been studied, as we discuss in the paper, artworks may be somewhat different due to their value in both as public hangings in a museum and as purchasable objects (see also for further discussion Berridge et al., 1998; 2009). Hence, liking and willingness-to-pay become very interesting factors in the art context. 

Based on this comment, we have revisited the Introduction (see pp. 3-7) and tried to better articulate the importance and interplay of these variables, particularly in regards to art objects. Please see also Discussion (p. 30) where we highlight the interesting knowledge gains of our research and for the field that may not be visible at first glance. we also added the additional references suggested by the Reviewer. Belke, B., Leder, H., & Augustin, D. (2006). Mastering style. Effects of explicit style-related information, art knowledge and affective state on appreciation of abstract paintings. Psychology Science, 48(2), 115; Silvia, P. J. (2005). Cognitive appraisals and interest in visual art: Exploring an appraisal theory of aesthetic emotions. Empirical studies of the arts, 23(2), 119-133. Hopefully these additions have better articulated the novelty and importance of this study. 

Similarly, the influence of expert opinions of the aesthetic liking of artworks is not a novel finding (see Kirk et al., 2009, as cited by the authors in the introduction). 

Reply: Thank you again. It is true that the influence of experts’ opinions on choices and ratings has also been investigated. We also refer in the manuscript to several papers (e.g., Belke et al., 2010; Kirk et al. 2011: Leder & Schwarz, 2017), which studied expert influence on choices/ratings, such as the work by Kirk and colleagues on this topic (2009,2011). However, experts’ influence in conjunction with social reputation and explicitly our variables (liking/willingness to pay, monetary/artistic value, private/public and different social reputational groups) has, to the best of our knowledge, not yet been addressed. Please see p. 5-6 where we have now attempted to better highlight this aspect. 

However, the use of hormonal data certainly adds an interesting element to the research. However, the rationale for the inclusion of the ‘Behavioral Activation and Anxiety Scale as Choice Predictors’ needed to be stronger. The ‘explorative inclusion’ of the LSAS anxiety scale appears overly speculative and it wasn’t clear what this added to the study.

Reply: Thank you for appreciating our attempt to include hormonal analysis, which is, in art research, rather new to the field. Considering the inclusion of the Behavioral Activation and Anxiety Scale as Choice Predictors, we want to first note that we referred to and justified at several places in the manuscript the use of the BIS/BAS and LSAS scale as supplementary measures in conjunctions with hormonal investigations and intervention studies (see, e.g., Dapprich, et al., 2021, Kutlikova et al., 2020; Losiak et al., 2016). However, considering the comment also from R2 to rephrase the term psychometrics to motivational factors (see remark and reply to R2 below), we have added an additional explanation in the Theoretical Background (see Scales of Motivational Factors, pp. 13) and Discussion (see Results of Motivational Factors, pp. 34-35). 

We also used the opportunity to add some more references especially considering the inclusion of the LSAS anxiety scale. Dapprich, A. L., Lange, W. G., von Borries, A. K. L., Volman, I., Figner, B., & Roelofs, K. (2021). The role of psychopathic traits, social anxiety and cortisol in social approach avoidance tendencies. Psychoneuroendocrinology, 128, 105207. https://doi.org/10.1016/j.psyneuen.2021.105207; Losiak, W., Blaut, A., Klosowska, J., & Slowik, N. (2016). Social anxiety, affect, cortisol response and performance on a speech task. Psychopathology, 49(1), 24-30. https://doi.org/10.1159/000441503

A major concern is the design and task that was used. An item’s position has an influence on whether it will be chosen, with there being a bias/preference to choose middle options (Valenzuela & Raghubir, 2009; Atalay et al. 2012) and it is often important to control for effects of position when a choice is made from similar options. Unfortunately in this design the effect of position was not controlled for because the middle artwork always had an artistic value and price that was in the middle, and so it is not possible to disambiguate the effect of position from the effect of the artwork’s attributes (artistic/price) on the liking/willingness-to-pay choices. The Means from table 1 are all close to the neutral value (2). Does this indicate frequent choice of the middle artwork? As the authors note, these means show that there was not much effect of condition. In relation to the effect of position how often was each artwork (in each position) chosen? This information does not seem to be included in the tables provided.

Reply: Thank you for pointing out this potential confound. Fortunately, while it is true that the average choice is close to the middle one, this does not mean that the mid-choice was the most frequently selected. The reason for this is that participants chose the two sides roughly to the same extent, which averages out, so that the mean is close to the middle option. In fact, there was a general avoidance of the middle position. Presumably, the manipulation of high artistic/monetary value was strong enough to induce a small bias against the mid-position. We added two figures in the Supplementary Materials that show the distribution of choices by location for both position (Figure S1A) and value position (Figure S1B) and reported this in the Results section (see Behavioral Results—Artwork Choices in Liking and Willingness-to-Pay, p. 21-23)

It is also not clear from the manuscript why a middle option was used in the design. It is not clear from the description what it added and how it enabled a comparison of the key variables of monetary value/aesthetic quality, when the left/right options already did this.

Reply: The decision to also include a middle option was due to our focus on social reputation effects of context. We wanted to offer participants explicitly the possibility of a “fallback” (i.e., middle/neutral) option, thus choosing neither the high monetary nor the high artistic value. This might present a form of flight or non-consent to either of the two groups of experts. Another possibility is that a person presents a change in choice in the public condition. We have updated the ‘Present Study’ section (see p. 8) to hopefully better make these arguments.

Please also see an addition to the Discussion (see p. 30-35) where we suggest that this is a highly interesting implication that should be closely examined in future studies, specifically the motivational effects and reasons for change in choice behavior. However, focusing on this are a matter of future studies. Addressing also this aspect would have been too much for adequately including in our study. Our approach was to deliver and present a first basis for this multimethod research: whether we can detect differences in choice behavior or not and whether we can influence them. This is also what we found. Our study hereby presents a first basis and legitimation to advance empirical research regarding the complex interplay of the addressed variables, value systems, and choice behavior considering aesthetic objects and art. 

In addition, there is substantial evidence that participants have a bias to look at the middle item (e.g. Tatler, 2007) and the eye tracking results show this expected middle looking bias in terms of more looks and last looks. The literature on the middle position and choice, and gaze behaviour, is not referred to and should be included (e.g. Atalay et al., 2012; van der Laan, et al., 2015). The possible impact of these effects on the results should also be considered in detail. In particular, it is possible that the gaze results predicted choice because the participants were looking more at the middle artwork. That is, was it an effect of position rather than artwork?

Reply: We would like to thank you very much for the detailed additions to the literature on eye tracking measurements, which we had not yet included. Thank you also for all feedback considering position effects, as this gave us the opportunity to include a more in-detail discussion of landing position and most fixations in the center. As described in the Methods (see section Stimuli, p. 15-16) the artwork positions were arranged randomly between participants. As now presented and added in the Supplementary Materials (see Figure S1) the artworks in the middle position, however, were not the most chosen ones. This delivers very interesting implications regarding when gaze predicts choice and when gaze does not predict choice due to social contextual influences or value systems. We also highlighted this now more thoroughly in the Discussion (see p. 30-32). We also added in the Discussion that the effect of finding most fixations in the middle, are due to the study design and how the artworks were presented. However, as we focused our hypotheses on gaze behavior predicting choice, the effect of position is not that relevant to test the hypotheses for the eye-tracking measures but must be seen in conjunction with the behavioral results. 

As suggested by the Reviewer, we also added the additional references: 

Atalay, A., Bodur, H., & Rasolofoarison, D. (2012). Shining in the center: Central gaze cascade effect on product choice. Journal of Consumer Research, 39(4), 848–866. https://doi.org/10.1086/665984 ; Tatler, B. W. (2007). The central fixation bias in scene viewing: Selecting an optimal viewing position independently of motor biases and image feature distributions. Journal of Vision, 7(14), 4. https://doi.org/10.1167/7.14.4; Valenzuela, A., & Raghubir, P. (2009). Position-based beliefs: The center-stage effect. Journal of Consumer Psychology, 19(2), 185–196. ; van der Laan, L., Hooge, I. T., de Ridder, D. T., Viergever, M., & Smeets, P. A. (2015). Do you like what you see? The role of first fixation and total fixation duration in consumer choice. Food Quality and Preference, 39, 46- 55. https://doi.org/10.1016/j.foodqual.2014.06.015.

The width of the artworks varied (height was kept constant), how were the authors sure that the participants were looking at a particular artwork, when the width varied from one trial to the next? How was the region of interest controlled?

Reply: The regions of interest were controlled for each triplet individually to cover the entire area to the outer edge of each artwork. It is true that the area of the individual ROIs varied, however, given the resolution of the eye tracker, we are quite confident in the ability to detect when individuals were or were not looking at the individual works of art. We have added further discussion on this point to the Methods section (see section Stimuli, p. 15-16). 

Why was a cluster analysis used for the hormonal analysis, rather than a regression? What was the rationale of creating two groups rather than treating hormone levels as a continuous variable?

Reply: We agree that categorization of continuous measures is not a perfect solution. However, one reason for this separation comes from the dual hormone hypothesis that suggests two groups in the hormonal data (as described on p. 11-13, see also Dekkers et al., 2019, Knight et al., 2019; Mehta & Prasad, 2015). We were thus heartened by the results that suggested that the patterns predicted by the theory also showed up in the clustering in a data-driven manner. 

A second reason for not adding the hormones as a continuous variable was that since the theory predicts that an interaction between testosterone and cortisol will affect behavior, and our behavioral result is also based on an interaction, we would have to look at the interaction of two interactions, for which our sample size is clearly not enough (as already a three way interaction can require 4 times as much data as a 2-way interaction). We therefore added further justification in the Measures and Statistical analysis section (see p. 20-21) and added the reference: Heo, M., & Leon, A. C. (2010). Sample sizes required to detect two-way and three-way interactions involving slope differences in mixed-effects linear models. Journal of biopharmaceutical statistics, 20(4), 787-802.

Participants were asked to make two choices on a trial: liking/willingness-to-pay. It is not clear from the information provided whether participants could choose the same artwork for both choices or had to choose a different artwork. The authors should include the task instructions for clarity.

Reply: Thank you very much for the comment. We have updated this in the section Procedures section (see pp. 16-17) and now hope to have provided a clearer indication of the operational sequence and choice options participants had. For your information, they were free to choose the same artwork or a different one in private compared to the public condition. This was important and of main value for the study design because we wanted to know if the social reputation influence in the public condition had an impact on the two choice types (meaning, if they would change their choice or not).

Fig S2 in supplementary Materials – did not seem to be included

Reply: We submitted the Figure (note new number due to the additional figures). We are sorry if this did not reach you and re-submitted now the Figure. 

Table 2 – Eye tracking analysis. Were these analyses corrected for multiple comparisons?

Reply: We report uncorrected p-values, we have updated the table and added a note to the adjusted p-value. (see Table 2, p. 25).

Line 154 – ‘group group’

Reply: Thank you very much. We corrected the double wording. 

Reviewer #2: 

The MS states in line 307 "Whereas representational paintings might be expected to have a higher, positive correlation between price and artistic quality". There is a debate about 'what' is artistic quality. As artistic quality is not a quality of the object but an individual assessment. I suggest to reframe the as perceived or assessed artistic quality.

Reply: Thank you very much for this comment about the term artistic quality itself, where we fully agree that this is an interesting and still undefined topic. Since our study, however, contained many values which needed to be defined and expressed, we did not want to leave the meaning of the term open for the participants nor lead participants to be busy with thinking about the potential different meanings of the values addressed. On the contrary, it was of utmost importance for the study to tie the values used as influencing variables to our main focus—social reputation. For this reason, a clear explanation of how to interpret the term “artistic quality” was given within the instructions on the computer screen. It was explained to the participants that the artistic quality is determined by experts of the “Academy of Arts” in Vienna with the explanation that the artistic value means that this work of art has, among experts, a special aesthetic, historical, scientific or social value for past, present and future generations. This statement has been provided in as full text in the Supplementary Materials and was explained in the Procedures. 

However, we thank you for the feedback and have now translated the text of the cover story in the Supplementary Materials additionally into English (see Supplementary Description D1 German version and D2 English version). We further added in the Discussion within the discussion of limitations the potential issue in conclusion of what for each participants artistic quality means (see Limitation, Emphasis, and Final Conclusion, p. 35-36).

In line 58, authors discuss about "scholarly discussions of judgment and taste". In my opinion, MS would improve with the addition of modern empirically based discussion of judgement and taste. There were even updates and criticisms of the Eysenck model of judgement cited in 19 (see 10.1016/j.paid.2017.05.041 , https://doi.org/10.1111/bjop.12427, https://doi.org/10.1111/bjop.12440, 10.1093/oxfordhb/9780198824350.013.40). Also, would be interesting to cite more recent views of neuroaesthetics and empirical aesthetics references (see https://doi.org/10.1016/j.cub.2018.06.004 and https://doi.org/10.1016/j.tics.2014.03.003)

Reply: Thank you again for your suggestion on the theoretical background and additional important references. First, we want to note that the excerpt was in the course of a brief historical overview of artistic and monetary values, mostly to call attention to the substantial importance of these values in art history, in art research, and within socio-cultural interaction. We therefore gave a literature selection up to the year 2017 (Pelowski et al., 2017). It was therefore not an exhaustive review as such but an introduction to the relevance of the terms and value systems along with such terms. We hope that the general reworking of the intro has made for a better line of argument. 

 In addition, we have now also included the references the Reviewer suggested considering the topic aesthetic sensitivity. Aesthetic sensitivity (defined as the ability to recognize and appreciate beauty and compositional excellence and to judge artistic value according to aesthetic standards, Corradi et al., 2020; Myshkowski et al., 2017, 2019, 2020) forms an essential part of the rationale for why we were able to draw on art objects for investigating choice behavior in our study design. We included this aspect in the Introduction (see p. 3-7) and Discussion (see p. 37). 

 Nonetheless, we would like to point out that our study does not focus on the aesthetic experience itself. We also did not study the ability to recognize aesthetic quality of objects, also referred to as aesthetic sensitivity (Corradi et al., 2019; Myszkowski et al., 2017, 2020). Rather, we investigated the decision-making behavior of liking and willingness-to-pay within a social reputational context using objects that are within the valuation context of aesthetics. Hence (and please also see our Reply and adjustments in the manuscript on the first part of the comment) we gave participants precise instructions for how to understand artistic quality within the experiment. Thank you again for pointing this out. Added references are: Corradi, G., Chuquichambi, E. G., Barrada, J. R., Clemente, A., & Nadal, M. (2020). A new conception of visual aesthetic sensitivity. British Journal of Psychology, 111(4), 630-658., https://doi.org/10.1111/bjop.12427; Myszkowski, N., & Storme, M. (2017). Measuring “good taste” with the visual aesthetic sensitivity test-revised (VAST-R). Personality and Individual Differences, 117, 91-100. https://doi.org/10.1016/j.paid.2017.05.041; Myszkowski, N., Çelik, P., & Storme, M. (2020). Commentary on Corradi et al.’s (2019) new conception of aesthetic sensitivity: Is the ability conception dead?. British Journal of Psychology, 111(4), 659-662. https://doi.org/10.1111/bjop.12440; Myszkowski, N. (2020) Aesthetic Sensitivity. In Nadal, M. & Vartanian, O.(Eds.), The Oxford Handbook of Empirical Aesthetics. https://doi.org/10.1093/oxfordhb/9780198824350.013.40

We also added the general references considering aesthetics and neuroaesthetics. Brielmann, A. A., & Pelli, D. G. (2018). Aesthetics. Current Biology, 28(16), R859-R863. https://doi.org/10.1016/j.cub.2018.06.004; Chatterjee, A., & Vartanian, O. (2014). Neuroaesthetics. Trends in cognitive sciences, 18(7), 370-375. https://doi.org/10.1016/j.tics.2014.03.003

In my opinion, the use of "personality" to refer to BIS and BAS could be misleading. Maybe, changing to "personality" to "individual differences" or traits could improve the clarity of the study.

Reply: We thank you very much for the suggestion. After consulting some colleagues in the field and re-evaluating the scale once more, we have come to the conclusion that the best term to use is “motivational factors”. The term has been corrected throughout the manuscript.

Also, in line with my previous comment, I think that referring to validated scales as "psychometric" is also misleading. Psychometrics usually refers to scientific study of psychological constructs measurement and assessment, while it is related to scale development, is not constrained to it.

Reply: In line with our reply above, we changed now the term psychometrics to scales of motivational factors. 

Authors state (line 394) that "Despite the discrete (1-2-3) choice options, once averaged across trials, the responses were normally distributed. This was confirmed by a Shapiro-Wills test, which found no deviation from normality (liking: W = .985, p = .197, wtp: W = .991, p = .591)". In general, I think that the analysis would benefit from linear mixed modelling as variation from participants and stimuli could be modelled (see 10.2190/6780-361T-3J83-04L1). Mixed models are specially suitable for empirical aesthetics research as stimuli (usually artworks) are a complex source of variability which not taken into account when using ANOVAs and t.test.

Reply: Thank you for the paper and the suggestion, we agree that mixed-models are in general well suited for empirical aesthetics (Silvia, 2007). We think that random slopes for each stimulus could be a suitable approach only if participants saw one stimulus at a time. However, in our design with 3 paintings presented on a trial (with different combination of paintings across participants), we did not find a standard model that would be suitable. We therefore did not include a mixed-model analysis. 

Also, I would suggest to update the current plots to a more informative ones with, at least, participant level information instead of point estimates with CI (see https://doi.org/10.1371/journal.pbio.1002128)

Reply: Thank you very much for this input. Based on your remark, we included some further figures and added them in the Supplementary Materials (see Figure S3 and S4). We also added the reference to the figures in the Results section (see Behavioral Results—Artwork Choices in Liking and Willingess-to-Pay, p. 213). We agree that a clear visualization of the between-participant variability is important and we therefore updated some of the plots. However, due to the large number of participants, we still think (also after seeing the new plots) that the averages demonstrate the underlying patterns in the data in a more comprehensible manner.

We also added both references in the according section: Silvia, P. J. (2007). An introduction to multilevel modeling for research on the psychology of art and creativity. Empirical studies of the arts, 25(1), 1-20. https://doi.org/10.2190/6780-361T-3J83-04L1; Weissgerber, T. L., Milic, N. M., Winham, S. J., & Garovic, V. D. (2015). Beyond bar and line graphs: time for a new data presentation paradigm. PLoS biology, 13(4), e1002128. https://doi.org/10.1371/journal.pbio.1002128

I value the effort of authors to be clear and honest about the exploratory stage of the study presented, the sensitivity analysis and the no a priori power determination. However I think that including the practical sample size determination (was budget?) would complete the "Participants" section.

Reply: Thank you for pointing out the missing explanation. We have added a paragraph in the Methods section (p. 14-15) that we wanted to reach at least 50 participants per between-group and in total over 100 peope based on the saliva sample measurement (Eisenegger et al., 2013). Nevertheless, due to the time and cost involved for conducting the study, it was not feasible to test even more participants at that time.

Regarding the eye tracking results, authors interpretation of data is „Studies have resoundingly shown that individuals spend more time looking at artworks which they find aesthetically appealing and which they liked most (in term of total fixation, see, e.g., [49, 51-54]; or along longer fixation, see [50]). Moreover, participants appear to fixate the preferred image when reaching the decision-moment [53-54].“ . But, there are other factors like complexity, interpretability, etc of the stimuli which could play a role (see 10.1177/0301006615596882, 10.1371/journal.pone.0037285). Maybe, taking into account other explanations could improve the MS.

Reply: Thank you very much for this remark. It has led us to substantially re-evaluate the Introduction and the Discussion to integrate further explanations. Especially your remark made us aware that we did not appropriately communicate that we were aware of these potentially influences and tried to address the obstacles for our study within the study design to the best of our knowledge. 

To come directly to your point, we agree that visual features like complexity, color, as well as expertise or interpretability, influence the decision process. We find especially the second paper by Massaro and colleagues (2012) exciting, investigating in their study the influence of (most likely) bottom-up visual features versus the influence of content-related top-down processes on aesthetic judgment. Interestingly, they also write that influences such as cultural background and learned judgment reference spaces have a significant impact on the evaluation of art choices or aesthetic judgments (see page 1, Massaro et al., 2012). In our Introduction as well as in the Discussion we explicitly address this influence and use it as a reference point of top-down mediated influences of socio-cultural learned associations on the decision-making process. We also want to mention here, that especially due to our focus on social reputational influences, we created a design around social reputation influences and socio-cultural learned value judgments. 

Nevertheless, we agree with you that we could have addressed and mention also important other influencing variables, such as low-level visual features, better (especially here referring to the seminal work of Daniel Berlyne starting in the 1960s, criticism made by Martindale et al., 1990, Marin et al., 2016, and even Chatterjee et al. 2010 work on the art attribute scale). These can make an artwork, for example, more salient and by that more recognizable or interesting for liking or buying interests. We were aware of this fact and therefore also explained in detail in the Stimuli section that we took great care to created triplets highly similar in style, color, composition, and also often painted by the same artists in the same period of carrier (by that also address art styles), so that our triplets appear most similarly (see in section Stimuli p. 15-16). This decision was taken because we wanted to focus the participants on the social reputational influences. This does certainly not exclude the influence of other factors then the ones we focused on. Hence, we addressed this in addition in the Discussion (see p. 30ff) and additional also in more detail now in the Stimuli description section (see p. 15-16). 

Do authors ensured that participants were naïve regarding art knowledge? Were any measure of it taken?

Reply: Yes. In the demographics we asked the participants about their art expertise with several general questions about educational background (see p. 19, see also Participants, p. 14). Our cohort were also only psychology students and they stated that they do not think that they are art experts themselves. Such questions are generally used in empirical art and aesthetic experiments (e.g., Leder et al., 2004; Leder & Schwarz, 2017). 

Note: I don't feel validated to assess the procedure, methods and results from the endocrinological part of the study.

Reply: We would like to thank you once again for the good remarks and openness towards our project and the Measures used, and appreciate especially your knowledge regarding the eye-tracking, behavioral measures and the terminology considerations.

---

## [Decision Letter · Decision Letter 1]

7 Mar 2022

PONE-D-21-26393R1

Social Reputation Influences on Liking and Willingness-to-Pay for Artworks: A Multimethod Design Investigating Choice Behavior along with Physiological Measures and Motivational Factors

PLOS ONE

Dear Dr. Spee,

Thank you for submitting your manuscript to PLOS ONE. After careful consideration, we feel that it has merit but does not fully meet PLOS ONE’s publication criteria as it currently stands. Therefore, we invite you to submit a revised version of the manuscript that addresses the points raised during the review process.

Since you have successfully incorporated all of the reviewers' comments, I will not further send this paper for review.

Reviewer 1 has pointed out a few minor changes that must be addressed. Concerning the comment on the possibility that position can affect the results, I agree with this reviewer in that this explanation cannot be ruled out completely. Thus, I encourage you to reword the corresponding sentence in the manuscript, to make it clear that this limitation exists.

Once these minor changes are made, I will be ready to accept the manuscript. Congratulations.

We look forward to receiving your revised manuscript.

Kind regards,

Fernando Blanco

Academic Editor

PLOS ONE

Journal Requirements:

Reviewers' comments:

Reviewer's Responses to Questions

**Comments to the Author**

1. If the authors have adequately addressed your comments raised in a previous round of review and you feel that this manuscript is now acceptable for publication, you may indicate that here to bypass the “Comments to the Author” section, enter your conflict of interest statement in the “Confidential to Editor” section, and submit your "Accept" recommendation.

Reviewer #1: (No Response)

Reviewer #2: All comments have been addressed

2. Is the manuscript technically sound, and do the data support the conclusions?

Reviewer #1: Yes

Reviewer #2: Yes

3. Has the statistical analysis been performed appropriately and rigorously? 

Reviewer #1: Yes

Reviewer #2: Yes

4. Have the authors made all data underlying the findings in their manuscript fully available?

Reviewer #1: No

Reviewer #2: Yes

5. Is the manuscript presented in an intelligible fashion and written in standard English?

Reviewer #1: Yes

Reviewer #2: Yes

6. Review Comments to the Author

Reviewer #1: I have now read the revised version of the manuscript. The authors have done a good job with the revisions and have largely addressed my concerns. The additional introductory information has provided a stronger rationale for the research and a fuller explanation of its novelty and importance.

Line 517-522. Although I am largely persuaded by the data (line 517-522) that position did not affect choice, I do not think it can be entirely discounted. Although the artworks were randomly allocated to position, the value labelling of an artwork in the left and right positions (Left, high-artistic: Right, high-monetary) remained the same throughout the task. Position was confounded with the value labelling of the artworks and ideally the position of the value labels should have been counterbalanced. This is relevant because, in addition to a preference for items in the center, there are also perceptual asymmetries, with a body of work finding that visual information on the left-side carries greater weight/importance when making a perceptual judgement (e.g. Nicholls et al. 1999, https://doi.org/10.1016/S0028-3932(98)00074-8). While I think the data suggest an effect of value labelling for particular positions may not have been a problem, it is not certain. Consequently, I think the statement (line 770-773) that ‘an effect of position could be ruled out’ is too strong. This should be attenuated to something like ‘The effect of position with respect to choice may be unlikely…’

Minor points

Line 143 – ‘greatly less’, should this be ‘less’.

Line 146, ‘have shown to be more vulnerability to reputation influences’ should be ‘…..more vulnerable….’

Line 765 “landing position, see for further reading [80, 97-99])”. Including the following reference in this list is appropriate as it examined the effects of position on the choice of artworks and also used triplets of similar artworks while measuring eye tracking (Kreplin et al. 2014, https://doi.org/10.1016/j.actpsy.2014.08.003).

Congratulations on an interesting study.

Reviewer #2: All the comments have been adressed. The authors did a good job.

7. PLOS authors have the option to publish the peer review history of their article (what does this mean?). If published, this will include your full peer review and any attached files.

Reviewer #1: No

Reviewer #2: **Yes: **Guido Corradi

---

## [Author Response · Author response to Decision Letter 1]

9 Mar 2022

Response to Reviewers & Editor

We are pleased that with our last revision we could address mostly all issues raised, included all the great feedback from the Editor and the Reviewers. We read, again, the manuscript thoroughly and corrected the minor changes suggested by Reviewer 1 (marked in red). 

Thank you very much for this great collaboration. We are looking forward to further working with you all towards a successful publication!

The authors

Reviewer #1: 

I have now read the revised version of the manuscript. The authors have done a good job with the revisions and have largely addressed my concerns. The additional introductory information has provided a stronger rationale for the research and a fuller explanation of its novelty and importance.

Reply: Thank you very much for reviewing the revised manuscript. We are pleased that our rationale could been strengthened within the revised manuscript, and we want to thank the reviewer for all the valuable feedback that has led to the new intro. We are also pleased that we have been able to address most of the issues to the reviewer’s satisfaction.

Line 517-522. Although I am largely persuaded by the data (line 517-522) that position did not affect choice, I do not think it can be entirely discounted. Although the artworks were randomly allocated to position, the value labelling of an artwork in the left and right positions (Left, high-artistic: Right, high-monetary) remained the same throughout the task. Position was confounded with the value labelling of the artworks and ideally the position of the value labels should have been counterbalanced. This is relevant because, in addition to a preference for items in the center, there are also perceptual asymmetries, with a body of work finding that visual information on the left-side carries greater weight/importance when making a perceptual judgement (e.g. Nicholls et al. 1999, https://doi.org/10.1016/S0028-3932(98)00074-8). While I think the data suggest an effect of value labelling for particular positions may not have been a problem, it is not certain. Consequently, I think the statement (line 770-773) that ‘an effect of position could be ruled out’ is too strong. This should be attenuated to something like ‘The effect of position with respect to choice may be unlikely…’

Reply: Thank you very much for reviewing this again. We corrected in line (now) 777-780 the wording. We also want to refer to the Methods section that the labelling was also counterbalanced between participants. Even the coin and brush sign (left/right) under each image was counterbalanced between participants. However, it is true, that labelling was not changed anymore within the same participant (we added this description in the text, see lines 386-389, and in the Figure, see line 203-208). The reasoning is that we did want the participants to remember the label weights, in order to avoid repetitive eye-movement patterns to the labels below the images. This would have been distractive for focusing on the choice task and for the eye-movement pattern analysis. 

Line 143 – ‘greatly less’, should this be ‘less’.

Reply: Thank you. We deleted the word ‘greatly’. 

Line 146, ‘have shown to be more vulnerability to reputation influences’ should be ‘…..more vulnerable….’

Reply: Corrected. 

Line 765 “landing position, see for further reading [80, 97-99])”. Including the following reference in this list is appropriate as it examined the effects of position on the choice of artworks and also used triplets of similar artworks while measuring eye tracking (Kreplin et al. 2014, https://doi.org/10.1016/j.actpsy.2014.08.003).

Reply: Thank you. We added the additional reference. 

Congratulations on an interesting study.

Reply: We would like to express our sincere appreciation for the review of our manuscript, the rich input, and feedback.

Reviewer #2: 

All the comments have been adressed. The authors did a good job.

Reply: We are very happy that we could address all the interesting points, suggestions, and issues raised made by the reviewer. We would like to thank the reviewer very much for all the effort, also checking the reviewed manuscript.

---

## [Editor Report · Decision Letter 2]

14 Mar 2022

Social Reputation Influences on Liking and Willingness-to-Pay for Artworks: A Multimethod Design Investigating Choice Behavior along with Physiological Measures and Motivational Factors

PONE-D-21-26393R2

Dear Dr. Spee,

We’re pleased to inform you that your manuscript has been judged scientifically suitable for publication and will be formally accepted for publication once it meets all outstanding technical requirements.

Kind regards,

Fernando Blanco

Academic Editor

PLOS ONE

Additional Editor Comments (optional):

-Please ensure that the figures are readable in the next technical-checking stages. In the current PDF they look blurry.

-Typos: "because both the paintings" (L778) to "because both paintings"
---

## [Editor Report · Acceptance letter]

30 Mar 2022

PONE-D-21-26393R2 

Social Reputation Influences on Liking and Willingness-to-Pay for Artworks: A Multimethod Design Investigating Choice Behavior along with Physiological Measures and Motivational Factors 

Dear Dr. Spee:

I'm pleased to inform you that your manuscript has been deemed suitable for publication in PLOS ONE. Congratulations! Your manuscript is now with our production department. 

Kind regards, 

on behalf of

Dr. Fernando Blanco 

Academic Editor

PLOS ONE